# Flow cytometry multiplexed method for the detection of neutralizing human antibodies to the native SARS-CoV-2 spike protein

Lydia Horndler[1], Pilar Delgado[1], David Abia[2], Ivaylo Balabanov[1], Pedro Martínez-Fleta[3], Georgina Cornish[4], Miguel A Llamas[5] iD, Sergio Serrano-Villar[6] iD, Francisco Sánchez-Madrid[3], Manuel Fresno[1], Hisse M van Santen[1] iD & Balbino Alarcón[1,*] iD

## Abstract

A correct identification of seropositive individuals for the severe acute respiratory syndrome coronavirus-2 (SARS-CoV-2) infection is of paramount relevance to assess the degree of protection of a human population to present and future outbreaks of the COVID-19 pandemic. We describe here a sensitive and quantitative flow cytometry method using the cytometer-friendly non-adherent Jurkat T-cell line that stably expresses the full-length native spike "S" protein of SARS-CoV-2 and a truncated form of the human EGFR that serves a normalizing role. S protein and huEGFRt coding sequences are separated by a T2A self-cleaving sequence, allowing to accurately quantify the presence of anti-S immunoglobulins by calculating a score based on the ratio of fluorescence intensities obtained by double-staining with the test sera and anti-EGFR. The method allows to detect immune individuals regardless of the result of other serological tests or even repeated PCR monitoring. As examples of its use, we show that as much as 28% of the personnel working at the CBMSO in Madrid is already immune. Additionally, we show that anti-S antibodies with protective neutralizing activity are long-lasting and can be detected in sera 8 months after infection.

**Keywords** flow cytometry; method; S protein; SARS-CoV-2; seropositivity
**Subject Categories** Immunology; Methods & Resources; Microbiology, Virology & Host Pathogen Interaction

## Introduction

Severe acute respiratory syndrome coronavirus-2 (SARS-CoV-2) is the causative agent of the global pandemic COVID-191. Phylogenetic analysis of the full genome classifies SARS-CoV-2 as a Betacoronavirus subgenus Sarbecovirus, lineage B and is related to bat isolates of SARS-CoV and is 79% identical to the SARS virus causing a viral epidemic in 2002 (Lu *et al*, 2020). Like other coronaviruses, SARS-CoV-2 encodes for 16 non-structural proteins at the 5′ end of the genomic RNA and structural proteins spike (S), envelope (E), membrane (M), and nucleocapsid (N) at the 3′ end (Su *et al*, 2016). The spike S protein is responsible for binding to ACE2 in host cells which seems to be the main cellular receptor for the virus (Hoffmann *et al*, 2020). In native state, the S protein forms homotrimers and is composed of two fragments S1 and S2 that result from proteolytic cleavage of S upon ACE2 binding (Walls *et al*, 2020). The S1 fragment contains a central RBD sequence that is the actual ACE2-binding sequence and the target of neutralizing antibodies such as those described to neutralize SARS-CoV-1.

Diagnosis of active infection is currently carried out by PCR amplification of viral RNA comprising fragments of the N, E, S, and RdRP genes from biological samples taken from bronchoalveolar lavage (BAL), sputum, nasal swabs, pharyngeal swabs, and fibronchoscope brush biopsies, with the highest positive rate resulting from PCR of BAL samples (Venter & Richter, 2020). Positivity in the PCR test vanishes as the infection is resolved although viral RNA shedding and PCR positivity could take place even after there is no further production of infective virions and therefore possibility of transmission.

Unlike PCR-based tests, serological tests are not highly valuable to determine which individual has an active infection with SARS-CoV-2 but are key in epidemiology and Public Health policies since it can provide an estimate of what segment of a population has been infected with the virus and is likely to have acquired total or partial immunity against ongoing and future resurgences of the pandemics. Herd immunity achieved either by natural infection or as a consequence of vaccination is a goal for all health authorities in the world.

1 Centro de Biología Molecular Severo Ochoa, Consejo Superior de Investigaciones Científicas (CSIC), Universidad Autónoma de Madrid, Madrid, Spain
2 Bioinformatics Facility, Centro de Biología Molecular Severo Ochoa, Consejo Superior de Investigaciones Científicas (CSIC), Universidad Autónoma de Madrid, Madrid, Spain
3 Immunology Department, Hospital Universitario La Princesa, HS-IP, Madrid, Spain
4 The Francis Crick Institute, London, UK
5 EMPIREO Diagnóstico Molecular SL, Madrid, Spain
6 Hospital Universitario Ramón y Cajal, Universidad de Alcalá, IRYCIS, Madrid, Spain
*Corresponding author. Tel: +34 911964555; Fax: +34 911964420; E-mail: balarcon@cbm.csic.es

Seropositivity is usually established by detecting the presence of viral antigen-specific IgG or IgM in the serum of individuals using recombinant fragments of the S or N proteins and tests based on ELISA or lateral flow assay (Venter & Richter, 2020; Weissleder *et al*, 2020). A disadvantage of those tests is that neutralizing antibodies are not directed against the N protein and that recombinant fragments of S miss the quaternary structure of the S protein trimer, which is the native form of the spike protein in the viral envelope. Therefore, part of the neutralizing antibodies directed against the native S trimer could be missed in serological tests based on the expression of recombinant proteins.

Flow cytometry of cells that are transfected with vectors that express the S protein represents an attractive alternative strategy for serological tests. This strategy allows detection of antibodies against the native form of the S protein. Such system has been used by transient transfection of HEK293T human cells to analyze different cohorts of COVID-19 patients (Grzelak *et al*, 2020). We have developed here a flow cytometry serological test using stably transfected Jurkat, a human leukemic T-cell line, that co-expresses the native S protein of SARS-CoV-2 and a truncated form of the human EGFR that serves as a normalizer. The Jurkat-S system is amenable to standardization and can be used for multiplexed detection of human immunoglobulins of all isotypes in a single assay. Finally, we show that the system is superior to ELISA-based methods to detect sera of donors containing neutralizing antibodies.

## Results

We used a lentiviral vector to express the full spike "S" protein of SARS-CoV-2 followed by a truncated human EGFR protein (huEGFRt) linked by a T2A self-cleaving sequence in transduced cells (Fig 1A). This system allows expression of the two proteins from a monocistronic mRNA. We produced transducing supernatants to express the construct in the human leukemic cell line Jurkat. Taking advantage of the fact that after cleavage the T2A sequence remains attached to the N-terminal protein, we studied whether the S protein was expressed at the cell surface of the transduced Jurkat cells (from now on, Jurkat-S cells). To this end, cell surface membrane proteins of Jurkat-S cells were surface biotinylated followed by immunoprecipitation with anti-T2A, SDS–PAGE, and Western blot with streptavidin–peroxidase. This showed the presence of a polypeptide of ~ 120 kD when resolved by SDS–PAGE under reducing conditions in Jurkat-S cells but not in the parental cells (Fig 1B). Interestingly, larger sizes of the immune-reactive polypeptide were resolved in the absence of reducing agents under non-denaturing conditions (i.e., without boiling; Fig 1C). These data suggest that the S protein of SARS-CoV-2 could be expressed on Jurkat-S plasma membrane as native homotrimers. To determine whether the S protein expressed on Jurkat-S cells was functional, we analyzed if it could promote the formation of syncytia when Jurkat-S cells were co-incubated with the ACE2-expressing human hepatocarcinoma cell line HepG2. We labeled Jurkat-S cells with the green dye CFSE and HepG2 with the far red dye Cell Trace Far Red (CTFR). After overnight incubation, we detected a Jurkat-S dose-dependent formation of mixed cells that were not detected when HepG2 were incubated with parental Jurkat cells not expressing the S protein (Fig 1D). To determine whether the mixed cells labeled with CFSE and CTFR were syncytia and not just cell doublets, the dual color cell population was sorted and seeded on coverslips for confocal microscopy analysis. The sorted population was formed by large cells with multiple nuclei that were stained with CFSE and CTFR, demonstrating that they were syncytia formed by fusion of Jurkat-S and HepG2 cells (Fig 1E). These data indicated that the S protein expressed in Jurkat-S cells has fusogenic activity and therefore that it must be in a native conformation.

The Jurkat-S cells were analyzed using a flow cytometry (FC) assay, where staining with an anti-EGFR monoclonal antibody detects and quantitates expression of the huEGFRt construct alongside detection and quantitation of anti-S protein antibodies within sera from SARS-CoV-2-infected blood donors. Figure 1F shows Jurkat-S cells stained with anti-EGFR mAb and either a serum sample taken from an individual before the COVID-19 pandemics (pre-COVID-1 serum) or serum taken from an asymptomatic donor (donor #15) determined positive with multiple serological assays (Table EV1). Serum from donor #15 could strongly detect the S protein expressed on Jurkat-S cells, whereas pre-COVID-1 serum did not, proving this method to be valid for capture and detection of anti-spike antibodies present in sera from SARS-CoV-2-exposed individuals. This assay was highly sensitive and detected anti-S protein

**Figure 1. Jurkat-S cells expressing native SARS-CoV-2 spike S protein allow the detection of anti-S protein antibodies in human sera.**

A  Lentiviral construct used to permanently express S protein in Jurkat. The full-length mature S protein is preceded by a leader sequence of GM-CSF and followed by a T2A sequence which is followed by a tail-less truncated human EGFR construct.

B  Expression of S protein on the plasma membrane of Jurkat-S cells assessed by surface biotinylation and followed by immunoprecipitation with anti-T2A, SDS–PAGE under reducing conditions, and Western blot with streptavidin–peroxidase. Biotinylation of the parental Jurkat cells was carried out in parallel as a negative control. The arrow indicates the position of the reduced S protein.

C  Expression of the S protein in native form was assessed by surface biotinylation of Jurkat-S and Jurkat cells followed by immunoprecipitation with anti-T2A followed by SDS–PAGE under non-denaturing conditions (i.e., without reducing agents and without boiling). The nitrocellulose membrane was blotted with streptavidin–peroxidase. Arrows point at different oligomerization forms of the S protein.

D  Formation of syncytia between the CTFR-labeled ACE2[+] human cell line and the CFSE-labeled Jurkat-S cells was measured by flow cytometry by analyzing the percentage of cells that become double positive for CTFR and CFSE markers. The bar plot to the right shows the effect of different doses of HepG2 cells on the formation of syncytia with a fixed number of Jurkat. Parental Jurkat cells (not expressing S protein) are considered negative controls. Data represent the mean ± SD of triplicated datasets.

E  Formation of syncytia between the CTFR-labeled ACE2[+] human cell line and the CFSE-labeled Jurkat-S cells, in an experiment as in (D), was confirmed by confocal microscopy after sorting cells double positive for CTFR and CFSE. Nuclei were identified by DAPI staining. Red arrowhead in the DAPI image indicates a nucleus of Hep-G2 origin; the green arrowhead the nucleus of Jurkat-S origin.

F  Overlay plot of Jurkat-S cells that were incubated with anti-EGFR mAb conjugated to Bv421 and serum from either donor #15 or from a pre-COVID-19 donor and followed by a secondary anti-human IgG1 antibody conjugated to PE.

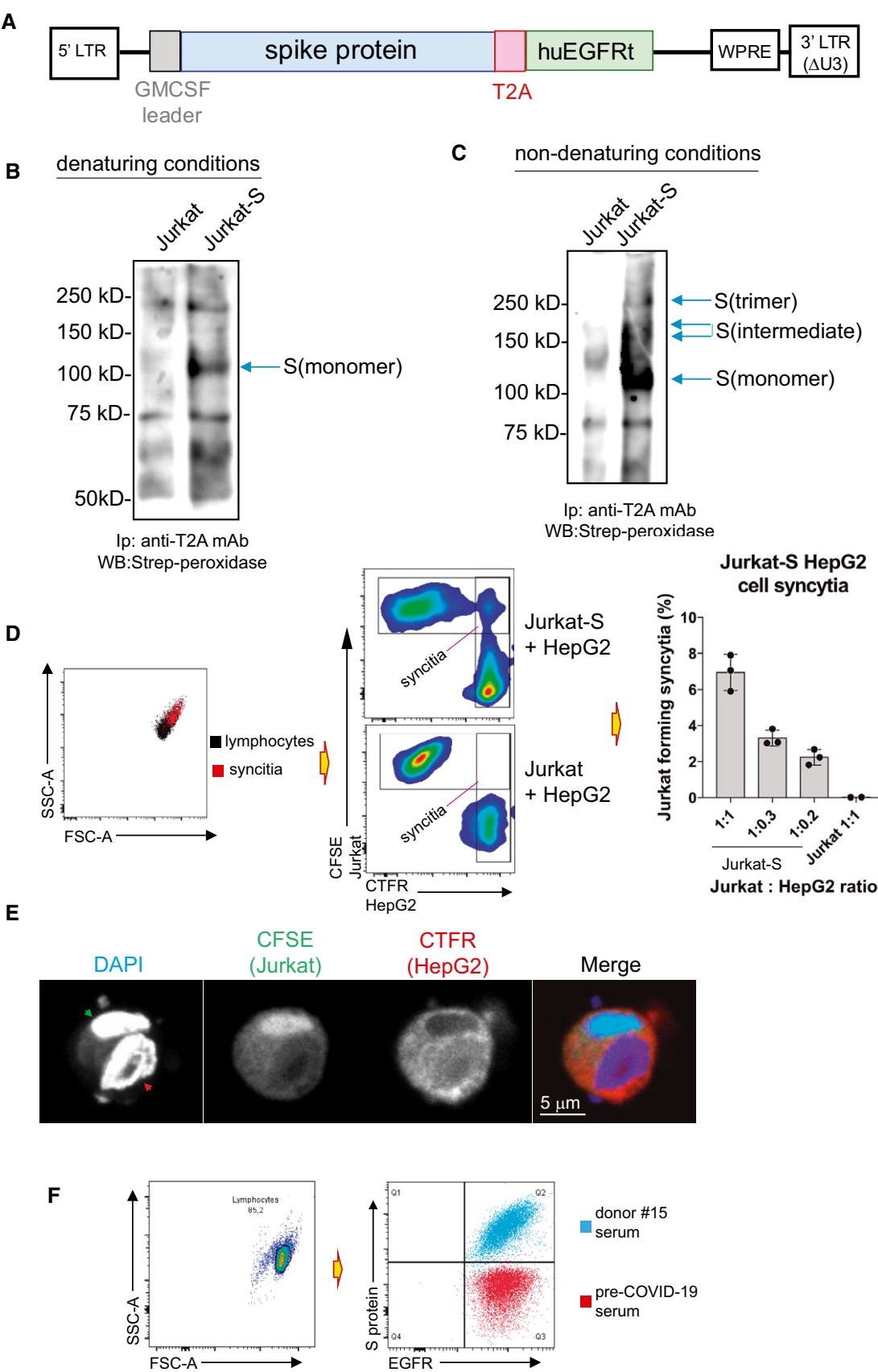

**Figure 1.**

antibodies over a wide range of mean fluorescence intensities (MFI) when staining with sera from different blood donors (Fig 2A). In comparison with commercial serological assays, this Jurkat-S clone FC assay detected more cases of seropositive individuals (Table EV1) and facilitated verification of positive seroconversion in patient samples that had previously given uncertain results (Fig 2B, orange symbols).

In order to compare the sensitivity of our Jurkat-S assay with a conventional ELISA, we used recombinant Spike proteins, specifically the S1 and receptor binding domain (RBD) fragments, both kind gifts from Dr. Peter Cherepanov (CRICK, London). We coated the plates with 2 µg/ml of either S1 or RBD and incubated the coated plates with a 1:50 dilution of human sera. Binding of human IgG1 antibodies was detected using a peroxidase-labeled anti-human IgG1 monoclonal antibody. Comparing absorbance values from the ELISA with MFI values of the Jurkat-S FC assay across sera stratified by experienced COVID-19 symptoms (asymptomatic, mild, moderate, moderate–severe, and severe) showed the MFI values to be spread across a wider range of values than absorbance values (Fig 3A). Finally, the comparison of absorbance values in the two ELISA tests (anti-S1 and anti-RBD) produced a good-fitted straight line ($R^2 = 0.83$; Fig 3B), whereas the comparison of the FC MFI with the absorbance values (against S1 and RBD) poorly adjust to a straight line ($R^2 = 0.55$ and $R^2 = 0.32$; Fig 3B), with samples that gave a poor signal in the ELISA giving a clear signal with the FC assay. This indicates that detecting S-specific IgG1 using the Jurkat-S FC assay increases sensitivity for detecting SARS-CoV-2-exposure in individuals testing negative by ELISA. These individuals may have generated antibodies against other fragments of the Spike, e.g. S2 (Ng *et al*, 2020), or against the native trimeric structure of the S protein, that are found on the surface of Jurkat-S cells but not in the ELISAs. For example, sera from donors #8, #46, #48, and #49 were apparently negative for anti-S IgG1 by ELISA but clearly positive by FC with Jurkat-S. By contrast, anti-S IgG1 was detected in sera from donors #15, #31, and #52 by both ELISA and FC, and serum from donor #58, appeared to be detected better by ELISA than by FC (Fig 3B). To determine whether those differences were maintained at different dilutions of the sera, a titration test was carried out in parallel using the FC Jurkat-S method and ELISA of the S1 fragment. The results showed that all sera, including that of donor #58, were clearly positive by FC, even at a 1:450 dilution, whereas by ELISA, sera #8, #46, #48, and #49 remained negative (Fig 3C). These results indicate that sera, which could have been considered negative by ELISA, are indeed positive for S-specific IgG1 by FC with Jurkat-S. Sample #49 was from an asymptomatic individual but sera #8, #46, and #48 were from individuals who had experienced symptoms that ranged from mild to severe (Table EV1).

To determine which of the two methods, ELISA or FC, was producing a more reliable picture of the immune status of the serum

donors, we assayed sera testing positive by FC and negative by ELISA for the capacity to neutralize S protein function. We generated pseudotyped lentiviral reporter particles coated either with the S protein of SARS-CoV-2 or, as a control, with the G protein of VSV virus (Fig 4A). To test the functionality of the viral particles, they were used to transduce HEK293T cells stably transfected with ACE2 in the presence of different dilutions of serum from donor #66, that was identified as positive by the FC method (Table EV1), o from the pre-COVID-1 donor. Serum #66 neutralized the entry of the S protein-pseudotyped lentivirus in a dose-dependent manner, whereas the pre-COVID-1 serum did not (Fig 4B). The effect of the #66 serum was due to specifically neutralizing the S protein since this serum did not inhibit transduction of ACE2$^+$ HEK293T cells by lentivirus pseudotyped with VSV G protein (Fig 4B). We then interrogated whether sera from donors #46 and #48, testing positive by FC but not by ELISA (Fig 3C), were also able to neutralize the S protein-pseudotyped lentivirus. They did inhibit (Fig 4C), suggesting that these serum samples contain neutralizing antibodies and therefore that the Jurkat-S FC assay can be superior to ELISA for detecting protective immunity to SARS-CoV-2.

To further determine the capacity of the FC Jurkat-S method to determine seropositivity in samples scored as negative by other methods, we tested serum samples collected between March and May 2020, from 30 healthcare workers (Hospital Ramón y Cajal, Madrid) that were repeatedly tested by PCR and ELISA for SARS-CoV-2 and determined to be negative in both tests (Table EV2). These samples were re-screened using the Jurkat-S FC assay and two sera (RyC52 and RyC65) resulted clearly positive for S-specific IgG1 (Fig 4D). To increase the sensitivity for detecting S-specific IgG1 in sera giving S/EGFR MFI ratios close to the 0.5 threshold for positivity, we plotted the EGFR MFI vs S protein MFI (Fig 4E) such as it was done in Fig 1F. S protein expression in Jurkat-S is coupled to huEGFRt expression (Fig 1A), thus positive sera produce a diagonal MFI on a plot of anti-S protein vs EGFR, whereas negative serum generates a flat profile. An example of this can be seen in Fig 4E with borderline sera RyC58 and RyC70. To further refine the method and establish a clear positive/negative threshold, we calculated an algorithm based on slope after fitting the distribution of anti-EGFR and anti-S fluorescence intensities to a Gaussian distribution (Materials and Methods and Fig EV1). The algorithm takes into account two mean points (the centers of the two Gaussian distributions) and the shape of the curves, defined by the co-variance matrixes (four parameters, Materials and Methods section). The result is a Score value with a threshold of 0.024 as the limit between positive and negative samples. Plotting the Score vs the S/EGFR MFI ratios for the 30 RyC samples results in a straight line (Fig 4F) and identifies samples RyC46 and RyC56 (in addition to the previously identified RyC52, RyC58, and RyC65 samples, Fig 4E) as positive. We carried out a neutralization assay with the S protein-pseudotyped lentivirus

---

**Figure 2.  Flow cytometry of Jurkat-S cells allows to detect anti-S immunoglobulins poorly detected by commercial tests with a wide dynamic range.**

A  Overlay histograms of Jurkat-S staining with different human sera diluted 1:50. A pre-COVID-19 serum sample is taken as negative control (gray histogram).
B  Bar plots of flow cytometry data generated with Jurkat-S cells and the panel of serum samples of Table EV1 classified according to their result in the indicated commercial tests: green, positive for the test; orange, weak, or unclear; magenta, negative samples. Flow cytometry data are expressed as the ratio between the MFI of the antibody anti-S and the MFI for EGFR. Negative values for the flow cytometry test are those with a S/EGFR MFI ratio lower than 0.5. This ratio was set in order to fit most of the data negative for the other serological tests (pink triangles) under that threshold. Data represent the mean ± s.e.m. All datapoints are shown.

**A**

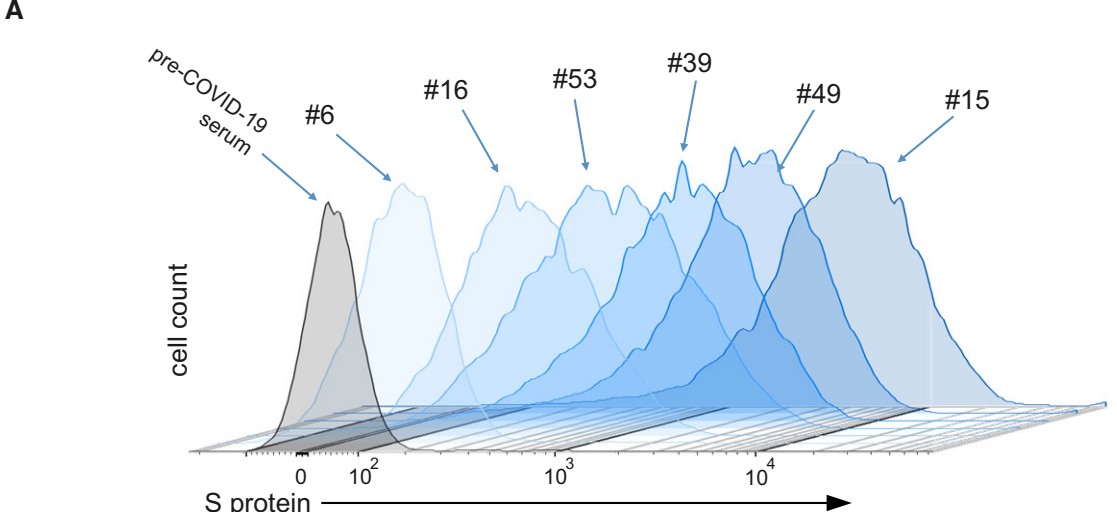

**B**

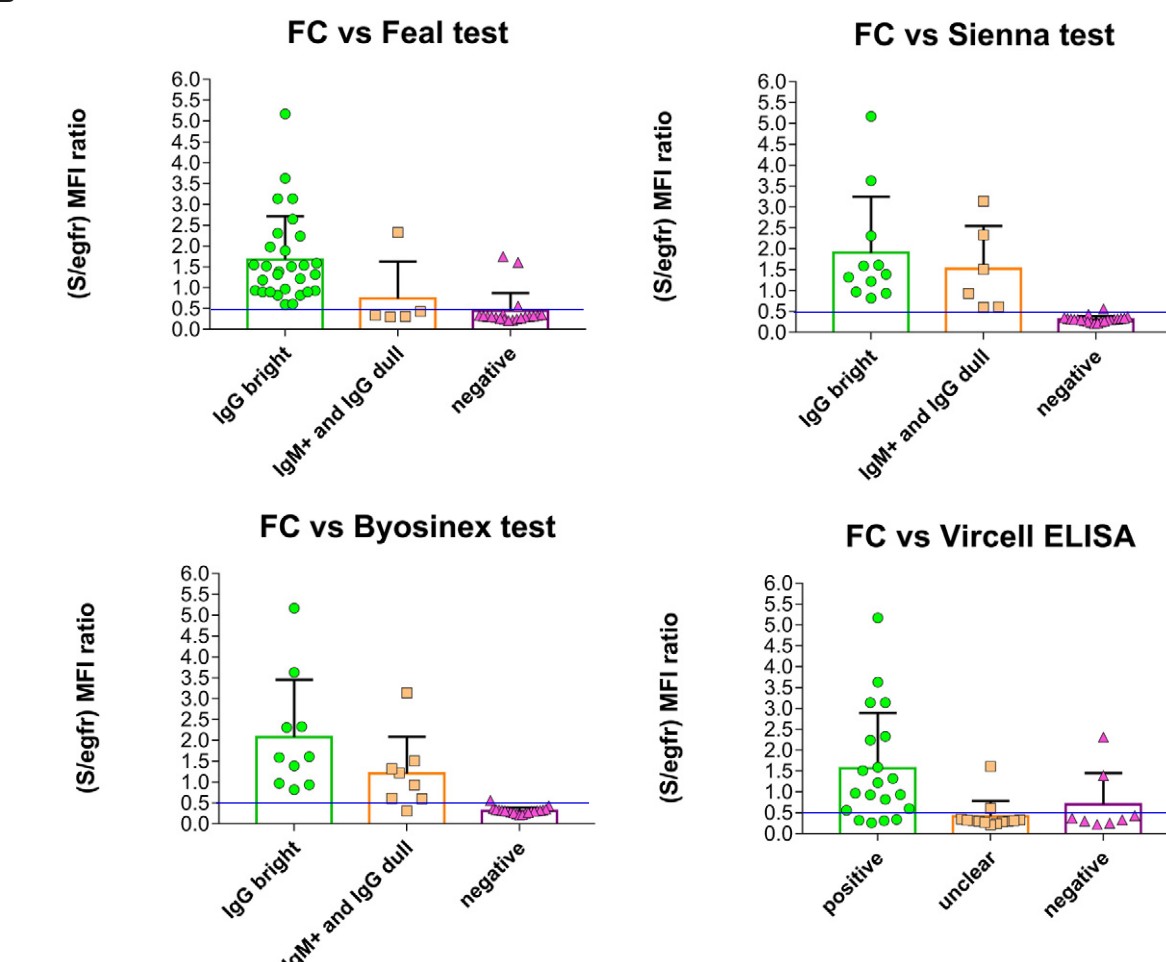

**Figure 2.**

and the available RyC samples that were positive by the FC method and found a weak, although significant, neutralizing activity for samples RyC46, RyC52, and RyC58 (Fig 4G), confirming the validity of the FC seropositivity data.

We moved onto multiplexing the Jurkat-S FC assay for the detection of a panel of anti-S immunoglobulins, including IgG1, IgG2, IgG3, IgG4, IgA, and IgM. We detected all anti-S isotypes in several serum samples and calculated the S/EGFR MFI ratio for each. The

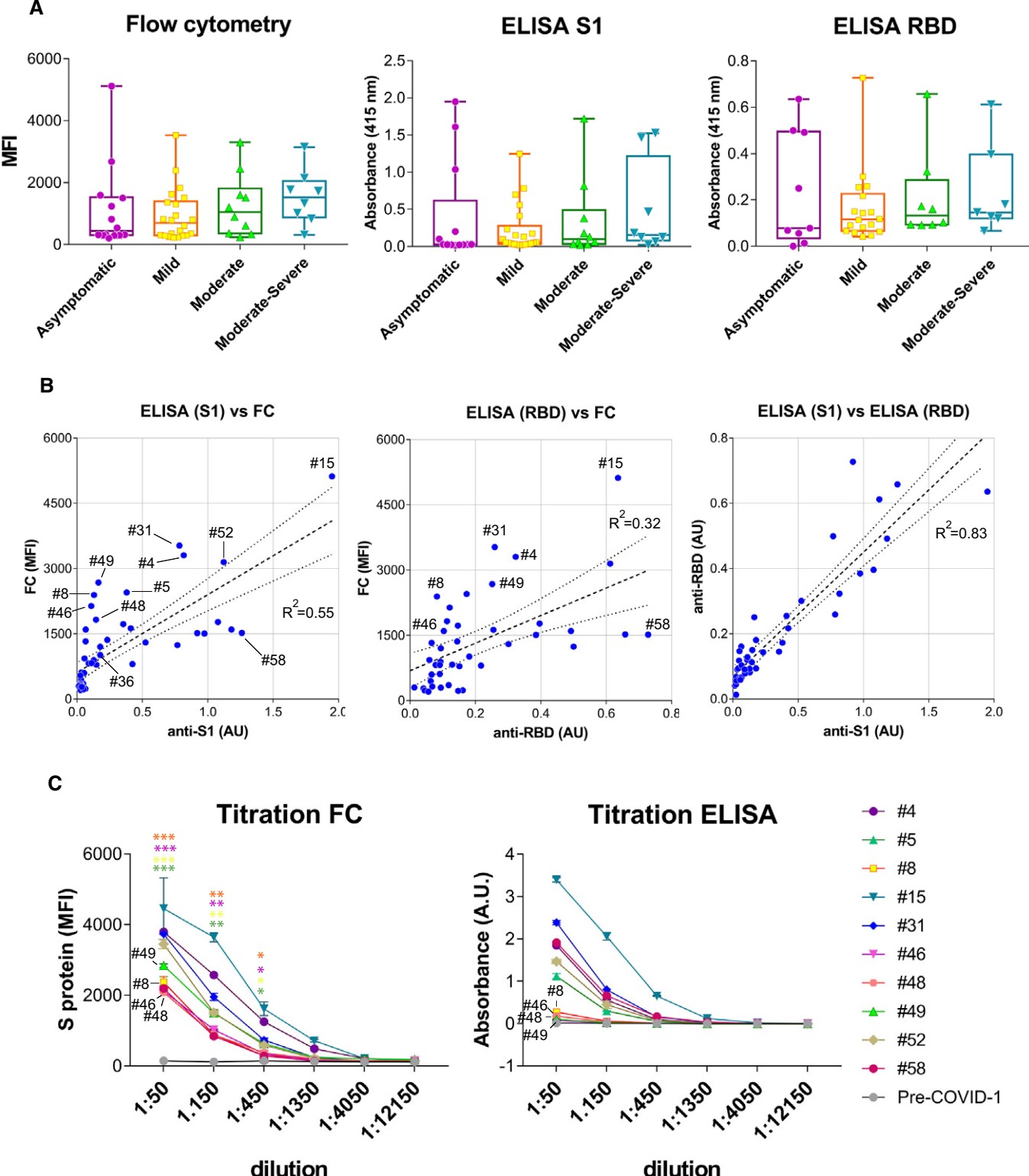

Figure 3.

**Figure 3.  Flow cytometry of Jurkat-S cells detects anti-S antibodies in sera otherwise negative for ELISA using recombinant S1 and RBD proteins.**

A   Box and whiskers plot of MFI and absorbance data in human sera according to their clinical classification of COVID-19 symptoms. Clinical classification was done according to the following parameters: Asymptomatic, no symptoms; mild, three or more of the following symptoms: non-productive cough, hyperthermia, headache, odynophagia, dyspnea, asthenia, myalgia, ageusia, anosmia, cutaneous involvement; moderate, three or more of the above symptoms plus gastrointestinal symptoms, or more than three of the above for seven or more days; and moderate–severe, three or more of the above symptoms plus pneumonia. Box and whiskers are shown to represent the minimum and maximum values as well as the median. All datapoints are shown.

B   Comparison of MFI vs absorbance data generated by flow cytometry and ELISA for all human sera. A lineal regression curve was adjusted, with a 95% confidence interval, to all data with the $R^2$ values indicated in the plots. Selected samples of outliers (#4, #5, #8, #31, #46, #48, #49, and #58) as well as samples close to a diagonal (#36, #52, and #15) are indicated.

C   Titration of selected human sera by ELISA using the S1 protein and by flow cytometry with Jurkat-S cells. Samples with antibodies detected by flow cytometry and not by ELISA are indicated (#8, #46, #48, #49). Data represent the mean ± SD of duplicates. An unpaired two-tailed $t$-test was carried out to compare the results of 1:50, 1:150, and 1:450 dilutions of sera #8, #46, #48 and #49 by FC to the pre-COVID-1 sample as a negative control. *$P < 0.05$; **$P < 0.005$; ***$P < 0.0005$.

most robust detection was observed for IgG1 in all cases, followed by IgG4 and IgA (Fig EV2A). A weak IgM response, in terms of S/egfr MFI ratio, was detected in all samples with the highest value in sample RyC65. The presence of anti-S antibodies of the IgG2 and IgG3 subclasses was not clearly detected in all samples. Figure EV2B shows a summary heatmap of the fold change in detection of each anti-S immunoglobulin isotype, referred to the MFI of the pre-COVID-19 sample. This indicates that the predominant humoral response to the S protein in the analyzed samples is in the form of IgG1 followed by IgM and IgG4.

To validate the FC Jurkat-S method, we decided to test 52 samples kept at the Hospital de la Princesa (HUP, Madrid), of which 40 had been determined positive by both PCR and an ELISA antibody method (Martínez-Fleta *et al*, 2020) (Table EV3). In addition, we included another set of 52 samples collected from donors before the onset of the COVID-19 pandemics in 2019. All HUP samples from PCR[+]ELISA[+] patients were also positive by the FC Jurkat-S method, giving a high Score (Fig 5A). By contrast, none of the pre-COVID sera gave positive results above the 0.024 threshold. These data show a high-level of correlation (> 97.5%) between samples of patients doubly identified as positive by PCR and a serological method, and our FC Jurkat-S method. In addition, no false-positive cases were identified in the cohort of 52 pre-COVID sera. The HUP cohort also revealed the presence of six samples tested negative by PCR that showed positive by both ELISA and the FC Jurkat-S method and one sample of serum from a PCR[+] patient that was negative by ELISA and the FC

Jurkat-S method (Fig 5A). We interpret those discrepancies as false PCR[+] for the latter case and false PCR negative for the six samples, probably because the samples for PCR were taken when the viral genome was no longer detectable (Table EV3). To determine which of the results was correct, if the PCR test or the serological tests, we chose four of the conflicting samples (Fig 5B) to be tested in the neutralization assay (Fig 4A). In spite of being negative by PCR, sera from donors HUP53, HUP56, and HUP60 neutralized the S protein-pseudotyped lentivirus (Fig 5C) indicating that those sera contained neutralizing antibodies, thus confirming the seropositive result using the FC Jurkat-S method. By contrast, serum from donor HUP59, who tested positive by PCR (Fig 5A and B), did not exhibit any neutralizing capacity, in agreement with the negative result in the FC Jurkat-S test. These results suggest that donor HUP59 was either a false PCR positive or was deficient in the development of a humoral response.

There is a controversy about the rapid decay of anti-SARS-CoV-2 antibodies, especially in persons with mild symptoms (Ibarrondo *et al*, 2020; Liu *et al*, 2020; Terpos *et al*, 2020; Wang *et al*, 2020). We assessed this issue by analyzing serum samples taken from eleven donors of the CBMSO with a variety of symptoms (Table EV4) in the months of June and October 2020, with a mean elapsed time of 123 days. We detected minor variations in the titer of anti-S antibodies by the FC Jurkat-S method with most donors not undergoing a significant decrease (Fig 6A). The titer in donor CBM-3 significantly increased in the June–October period suggesting the possibility of a reinfection with SARS-CoV-2. Although reactivity

**Figure 4.  Sera not detected by ELISA but detected by flow cytometry with Jurkat-S contain neutralizing antibodies.**

A   Cartoon of the strategy for generation of pseudotyped lentiviruses. HEK-293T cells were transfected with a lentiviral construct containing the EGFP marker gene and two more plasmids encoding for the gag and pol genes of HIV-1 and for the S protein of SARS-CoV-2. Alternatively, the latter plasmid was replaced by another one encoding the G protein of VSV. The cell supernatants collected after 48 h of transfection were used to transduce HEK-293T cells stably transfected with human ACE2.

B   Validation of the neutralization method was carried out with cell HEK-293T cell culture supernatants containing lentiviral particles pseudotyped with either the S protein or with the VSV G protein. The supernatants were mixed with sera from donor #66 at the indicated dilutions or from a pre-COVID sample as a control before addition onto HEK-293T-ACE2[+] cell cultures. No serum refers to a control containing no human sera. Data represent the mean ± SD of triplicates.

C   The presence of neutralizing antibodies in sera from donors #46 and #48 found negative by ELISA and positive by flow cytometry (Fig 3C) was demonstrated as in panel B. Data represent the mean ± SD of triplicates. *$P < 0.05$; **$P < 0.005$; ****$P < 0.00005$ (paired two-tailed $t$-test comparing all serum dilutions to the pre-COVID sample).

D   Bar plot showing the S/EGFR MFI ratio determined by flow cytometry analysis with Jurkat-S of serum samples from 30 sanitary personnel repeatedly tested as PCR negative for SARS-CoV-2 at the Ramón y Cajal Hospital of Madrid. A negative result is considered for a ratio lower than 0.5. Two clear cases of sera positive for anti-S IgG1 are indicated (RyC52 and RyC65). A borderline sample just above the threshold line (RyC58) is also indicated. Data show the mean ± s.e.m. All datapoints are shown.

E   Two-color dot plot overlay of anti-S protein fluorescence vs EGFR fluorescence for borderline serum sample RyC58 (blue) and the clearly negative RyC70 sample (red). The line plot to the right shows the MFI for EGFR taken at the three sectors indicated in the two-color plot and the corresponding MFI values for EGFR.

F   Plot of the S/EGFR MFI ratio for the 30 RyC samples vs a Score generated according to the slope of S/EGFR and shape (see Materials and Methods and Fig EV1). The green line indicates the threshold point (0.024) for the Score above which a serum is considered positive.

G   The presence of neutralizing antibodies in sera from donors RyC46, RyC52, RyC56, and RyC58 diluted 1/15 was demonstrated as in panel B. Data represent the mean ± SD of triplicates. *$P < 0.05$; **$P < 0.005$; ****$P < 0.00005$ (unpaired two-tailed $t$-test comparing each serum to the no serum sample).

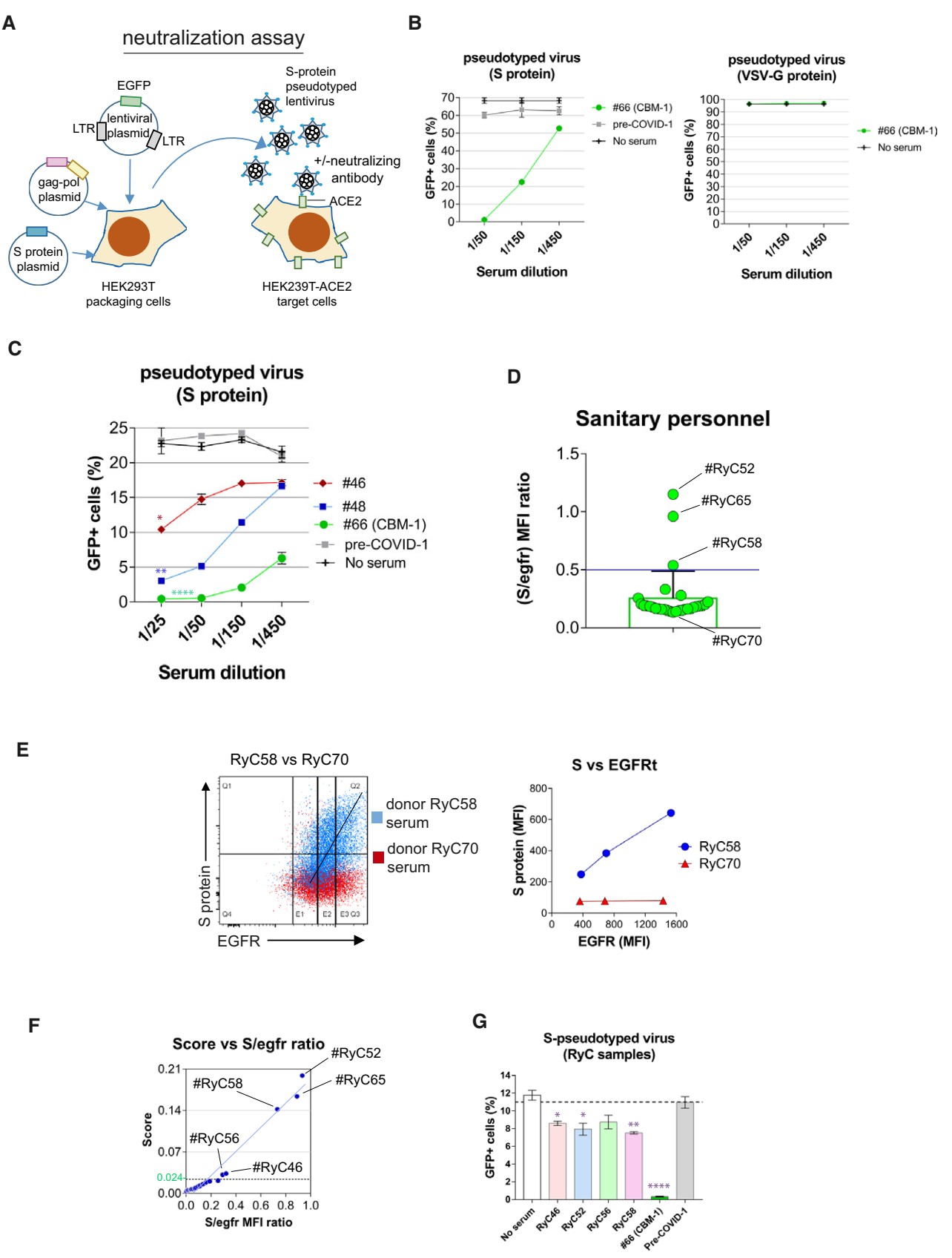

**Figure 4.**

**A**

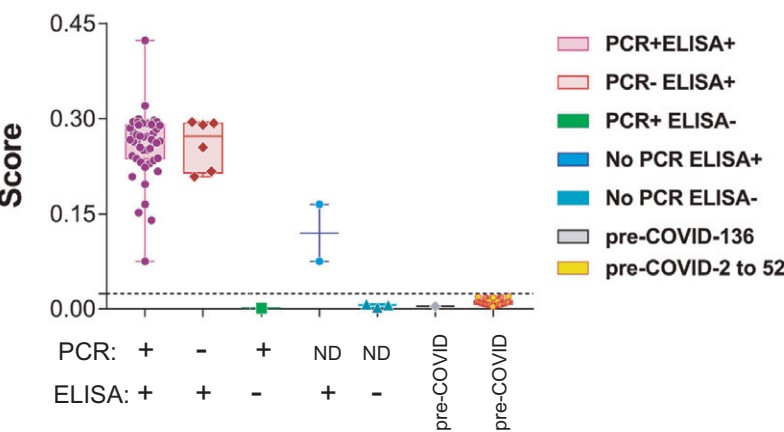

**B**

| Patient code | Age | Sex | PCR test | Result PCR | IgG anti-RBD ELISA | S/EGFR MFI ratio | Score | Result FC Jurkat-S test |
|---|---|---|---|---|---|---|---|---|
| **HUP53** | 53 | F | Yes | Negative | Positive | 0.78 | 0.45 | Positive |
| **HUP56** | 61 | M | Yes | Negative | Positive | 0.95 | 0.45 | Positive |
| **HUP59** | 28 | F | Yes | Positive | Negative | 0.18 | -0.12 | Negative |
| **HUP60** | 31 | F | Yes | Negative | Positive | 1.19 | 0.44 | Positive |

**C**

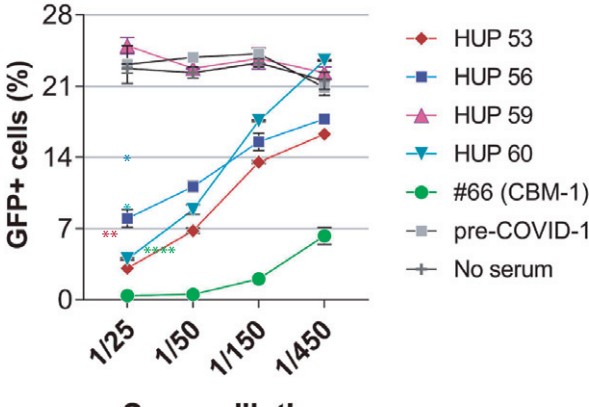

**Figure 5. Validation of the FC Jurkat-S method.**

A   A total of 52 serum samples from Hospital Universitario La Princesa (HUP) and 52 samples of pre-COVID donors stored at the CBMSO were tested by the flow cytometry method and classified according to the S/EGFR-based Score. Of the 52 HUP samples, a total of 40 were from donors testing positive by PCR and by an ELISA test, six were from donors testing negative by PCR but positive by ELISA, one from an individual testing negative by ELISA and positive by PCR, and five were not tested by PCR. Box and whiskers are shown to represent the minimum and maximum values as well as the median. All datapoints are shown. The broken line indicates the threshold for a Score value of 0.024.

B, C   Four HUP serum samples giving discrepant results by PCR and by the flow cytometry method were tested in the neutralization assay of pseudotyped lentivirus (Fig 4A). Data represent the mean ± SD of duplicates. *$P < 0.05$; **$P < 0.005$; ****$P < 0.00005$ (paired two-tailed *t*-test comparing all serum dilutions to the pre-COVID sample).

against S protein was still clearly detected in all samples taken in October, we decided to compare the neutralization capacity of the June and October samples of two of the donors with the highest loss in titers (CBM-1 and CBM-2 with a 2.3- and 2.0-fold loss, respectively, Fig 6A). We found a good correlation between the S/EGFR MFI ratio in the FC Jurkat-S assay and the neutralization capacity of the four serum samples (Fig 6B). The loss of anti-S antibody titer was inconsequential for donor CBM-1 and both the June and October samples equally neutralized entry of the S-pseudotyped virus up to a 1/4,000 dilution. Serum from donor CBM-2 experienced a small drop in neutralization capacity between June and October from a positivity in the neutralization test at a 1/150 dilution to a 1/50

dilution (Fig 6B). Nonetheless, those data indicate that anti-S antibodies are not decaying so fast and there is still neutralization capacity 8 months after the onset of symptoms in February 2020 (donor CBM-10, Table EV4).

In an attempt to use the FC Jurkat-S method to determine the existence of humoral immunity to SARS-CoV-2, we analyzed a cohort of serum samples from 415 volunteers working at the CBMSO and collected between June and October 2020. Some of them had been tested by PCR and/or by ELISA tests showing discrepancies with our FC Jurkat-S test (Fig 7A). Using our neutralization assay with S protein-pseudotyped virus, we found that all samples determined positive in the FC test contained neutralizing

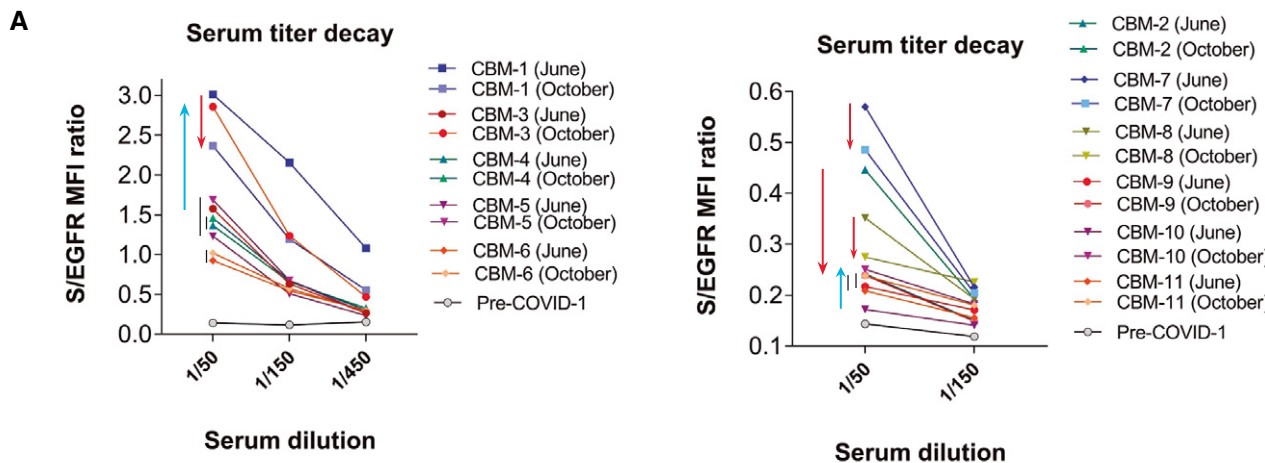

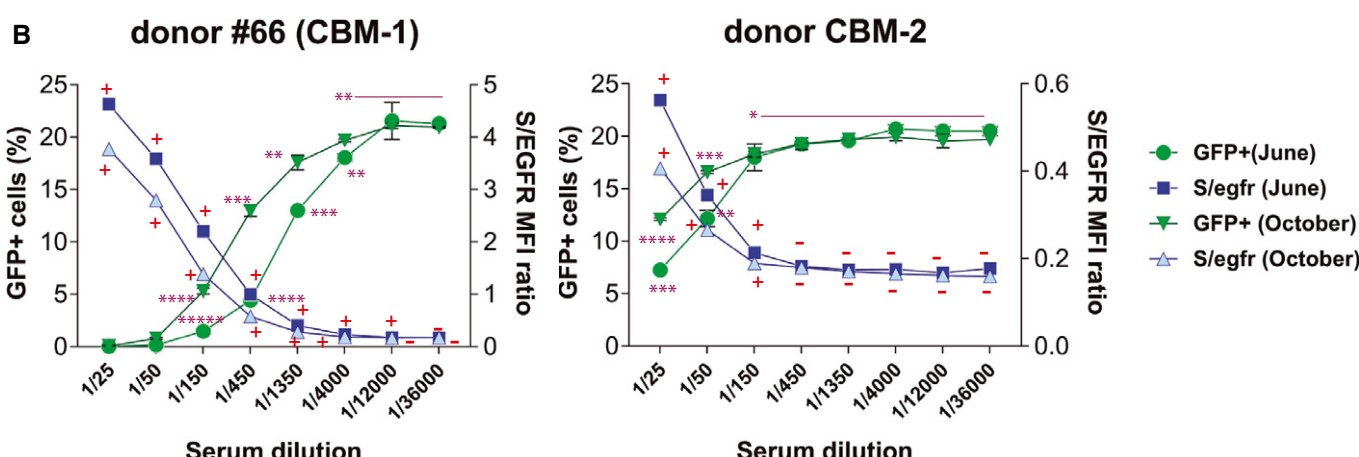

**Figure 6. Evolution of the presence of anti-S antibodies between the months of June and October 2020.**

A A collection of serum samples from volunteers working at the CBMSO were taken in the months of June and October 2020 and tested by flow cytometry using Jurkat-S cells at the indicated dilutions. Flow cytometry data are shown as the S/EGFR MFI ratio. Red arrows indicate a loss in titer of antibodies between June and October, whereas blue arrows indicate an increase in antibody titer.

B Titration of the neutralizing capacity of sera from a high titer (#66-CBM-1) and a low titer (CBM-2) donor taken in June and October 2020. The neutralization experiment was carried out in parallel to a FC Jurkat-S test. Flow cytometry results are shown as the S/EGFR MFI ratio (right y-axis) and according to the Score value. A plus symbol (+) indicates a Score > 0.024 for a given serum dilution; a minus symbol (−) indicates a Score < 0.024. Neutralization data are shown as the percentage of HEK-293T-ACE2$^+$ cells becoming GFP$^+$ (left y-axis). Data represent the mean ± SD of duplicates. *$P < 0.05$; **$P < 0.005$; ***$P < 0.0005$; ****$P < 0.00005$; *****$P < 0.000005$ (unpaired two-tailed t-test).

**A**

| Patient code | Age | Sex | PCR test | Result PCR | ELISA | S/EGFR MFI ratio | Score | Result FC Jurkat-S test |
|---|---|---|---|---|---|---|---|---|
| CBM-6 | 36 | F | No | | | 2.20 | 0.34 | Positive |
| CBM-11 | 52 | F | Yes | Negative | | 1.11 | 0.28 | Positive |
| CBM-12 | 62 | F | No | | Negative | 3.58 | 0.34 | Positive |
| CBM-13 | 28 | M | Yes | Positive | Negative | 0.96 | 0.42 | Positive |
| CBM-14 | 27 | M | Yes | Positive | Negative | 0.96 | 0.24 | Positive |
| CBM-15 | 25 | F | Yes | Positive | Negative | 1.45 | 0.38 | Positive |

**B**

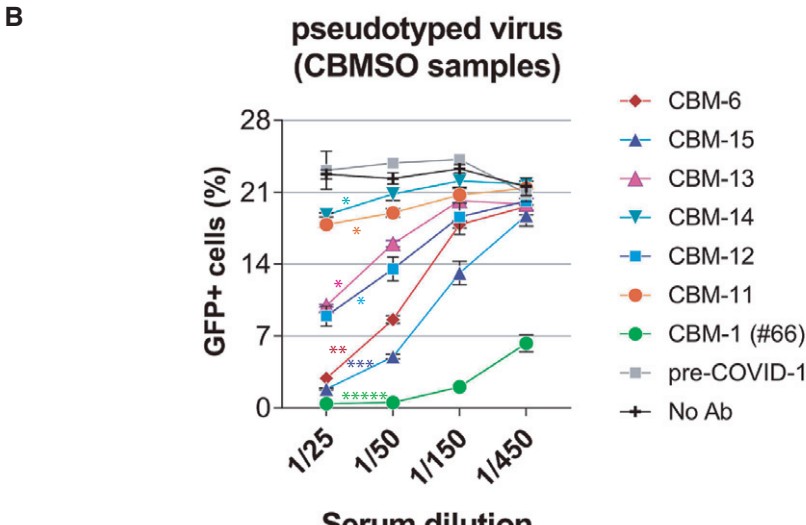

**C**

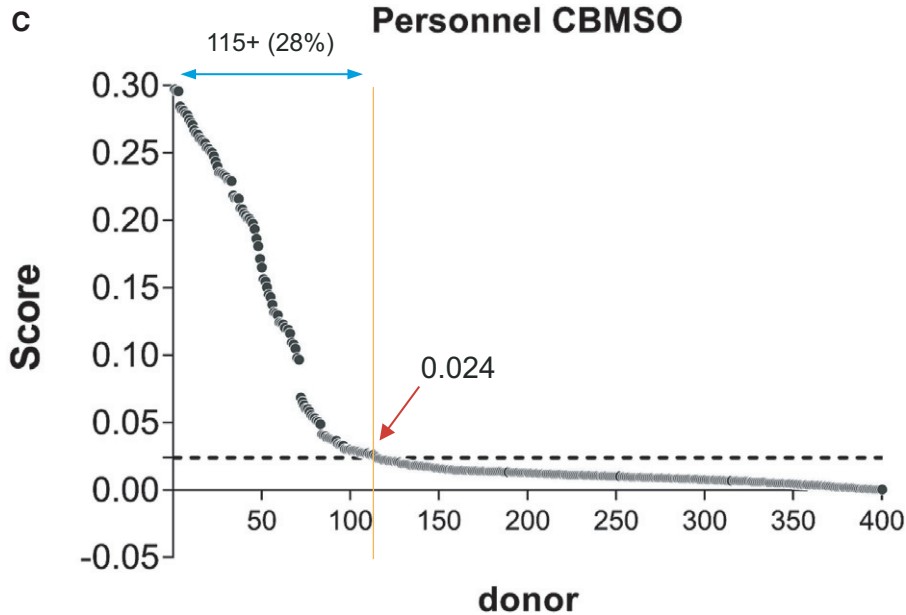

**Figure 7. Prevalence of seropositivity by the FC Jurkat-S method within a cohort of 415 samples.**

A   Discrepancies between the result of the flow cytometry test and previous PCR or ELISA tests within the CBMSO cohort.

B   A neutralization assay with S protein-pseudotyped lentivirus was carried out to resolve the discrepancies. Data represent the mean ± SD of duplicates. *$P < 0.05$; **$P < 0.005$; ***$P < 0.0005$; *****$P < 0.000005$ (paired two-tailed *t*-test comparing all serum dilutions to the pre-COVID sample).

C   Classification of the CBMSO cohort according to the Score generated from S/EGFR data. The broken line indicates the 0.024 threshold above which samples are considered positive.

antibodies (Fig 7B), indicating that the results of our FC test were correct. Of the 415 samples tested, a total of 115 resulted with a positive Score above the threshold of 0.024 (Fig 7C), indicating that a 28% of the analyzed CBMSO population has developed humoral immunity.

## Discussion

Here, we describe a method based on flow cytometry of a human T lymphoblastic leukemia cell line that stably expresses the S protein of SARS-CoV-2. Compared to methods based on the use of recombinant purified proteins, the FC method with Jurkat-S offers a wider range of response and it is highly quantitative. The coordinated expression of huEGFRt serves the function of a normalization marker and of unambiguously determining the positivity of sera with low antibody titers. The system can also be used in a multiplexed format to simultaneously detect all immunoglobulin subclasses specific for the S protein. Importantly, it is superior to current ELISA methods to determine the fraction of the human population that has acquired antibody responses and neutralizing activity toward SARS-CoV-2.

A FC-based method has been previously described using HEK293T cells that overexpress the S protein (Grzelak et al, 2020; Ng et al, 2020). Compared to this, the Jurkat-S system described here offers the advantage of employing a non-adherent cell line that does not require methods such as trypsinization to place them in suspension. In addition, as said above, the coordinated expression of the huEGFRt marker from a single polypeptide makes it very convenient to resolve ambiguous samples. This property has served to detect as seropositive 5 cases out of a cohort of 30 sanitary personnel highly exposed to SARS-CoV-2 and repeatedly considered as negative by PCR and other serological methods. The method has been validated using samples from contrasted PCR[+] and seropositive patients as well as sera collected before 2019. In all cases in which there has been a discrepancy between the FC Jurkat-S method and other serological methods, the FC method has been backed by neutralization data using S protein-pseudotyped lentiviral particles. The FC Jurkat-S method detects as seropositive serum samples that actually contain neutralizing antibodies.

To ascertain the usefulness of the FC method, we have measured the presence of anti-S antibodies in paired samples of sera taken from the same donors within a 4-month interval. We found that the titer of antibodies is in general stable during the studied period and remain as long as 8 months after the reported exposure to SARS-CoV-2. Furthermore, after 8 months of exposure to the virus, sera are shown to contain neutralizing immunity. Thus in the controversy about the duration of effective humoral response (Ibarrondo et al, 2020; Liu et al, 2020; Terpos et al, 2020; Wang et al, 2020), our results support that anti-S functional neutralizing antibodies are long-lasting.

In another example, we have used the FC Jurkat-S method to study the seroprevalence in a cohort of 415 individuals working at the CBMSO in Madrid with samples collected between June and October 2020. Our data show that 28% of the tested sera is positive for antibodies against the S protein, a percentage that is much higher than that of other studies carried out with samples of Madrid using other serological methods (Pollán et al, 2020). It would be interesting to use the FC method in the coming months

after the onset of the second wave of the pandemics and potentially obtain a more accurate determination of seroprevalence within the general population. Another conclusion that can be drawn from the analysis of the Empireo and HUP cohorts (Tables EV1 and EV3) by the FC Jurkat-S method is that the humoral response to the S protein is not inferior for the older donors (ages 60–70 and more) than for younger ones, suggesting that age "per se" is not conditioning the response.

Compared to ELISA and rapid serological tests, the FC Jurkat-S method has the disadvantage of requiring the use of a flow cytometer. However, the extended use of those machines in the immunology and hematology departments of hospitals may make the method of choice if the goal is to accurately determine what proportion of a human population likely has a protective humoral response and how far is from reaching the anxiously sought herd immunity. In summary, we believe that this method may represent an excellent addition to COVID-19 Epidemiology and Public Health policies.

## Materials and Methods

### Cells

The human T-cell line Jurkat clone E6-1 was acquired from ATCC (TIB-152) and was maintained in complete RPMI 1640 supplemented with 5% fetal bovine serum (FBS, Sigma) in a 5% $CO_2$ incubator. Human embryonic kidney HEK293T cells (ATCC CRL-3216) and human hepatocellular carcinoma HepG2 cells (ATCC HB-8065) were maintained in DMEM (HEK293T) or RPMI (HepG2) supplemented with 10% FBS in a 5% $CO_2$ incubator. HEK293T cells expressing ACE2 were generated by lentiviral transduction with vector CSIB and selection in blasticidin S (Robbiani et al, 2020) All cell lines were routinely tested for the absence of mycoplasma.

### Lentiviral vector and Jurkat cell transduction

To express the full-length spike S protein of SARS-CoV-2, we used the lentiviral vector based on the epHIV-7 plasmid that contains the truncated version of human EGFR (huEGFRt) that lacks the domains essential for ligand binding and tyrosine kinase activity described in Wang et al (2011).

For transduction, lentiviral-transducing supernatants were produced from transfected packaging HEK-293T cells as described previously (Martinez-Martin et al, 2009). Briefly, lentiviruses were obtained by co-transfecting plasmids pCMV-dR (gag/pol) and using the JetPEI transfection reagent (Polyplus Transfection). Viral supernatants were obtained after 24 and 48 h of transfection. Polybrene (8 µg/ml) was added to the viral supernatants prior to transduction of Jurkat cells. A total of $3 \times 10^5$ Jurkat cells were plated in a P24 flat-bottom well in 350 µl of RPMI, and 350 µl of viral supernatant were added. Cells were centrifuged for 90 min at 1,600 g and 32°C and left in culture for 24 h. Transduced cells were selected by FACS sorting 48 h later using the anti-EGFR antibody.

### Human sera

A total of 54 human sera were obtained from volunteers that contacted EMPIREO SL (www.empireo.es) for serological tests.

Serum donors filled in a questionnaire to allow their clinical classification according to the following parameters: asymptomatic, no symptoms; mild, three or more of the following symptoms: nonproductive cough, hyperthermia, headache, odynophagia, dyspnea, asthenia, myalgia, ageusia, anosmia, cutaneous involvement; moderate, three or more of the above symptoms plus gastrointestinal symptoms, or more than three of the above for seven or more days; moderate–severe, three or more of the above symptoms plus pneumonia; and severe, pneumonia requiring hospitalization and intubation. A second set of 30 serum samples was obtained from healthcare workers at the Intensive Care Unit of the Ramón y Cajal Hospital in Madrid. Those workers have been routinely tested for COVID-19 by PCR and resulted always negative. A third cohort of 52 serum samples was selected from the study "Immune response dynamics as predictor of COViD-19 disease evolution. Implications for therapeutic decision-making" from the Hospital Universitario La Princesa approved by the Research Ethics Committee (no. #4070). Of those, 40 sera samples were from COVID-19 patients diagnosed by PCR. All 52 serum samples were previously screened with an ELISA test for the SARS-CoV-2 RBD fragment of the S protein (Martínez-Fleta *et al*, 2020). A fourth set of 52 serum samples were collected between 2010 and 2018, before the COVID-19 pandemic and stored at −80°C at the CBMSO. Those samples constitute the pre-COVID group. Finally, a fifth cohort of 415 serum samples was obtained from capillary blood of volunteers working at the CBMSO in the period of June–October 2020. All participants provided written consent to participate in the study which was performed according to the EU guidelines and following the ethical principles of the Declaration of Helsinki.

### Flow cytometry

Jurkat-S cells were incubated for 30 min on ice with 1:50 dilutions of human sera in phosphate-buffered saline (PBS), 1% bovine serum albumin (BSA), and 0.02% sodium azide. Cells were spun for 5 min at 900 *g*, and the pellet was resuspended in PBS-BSA buffer and spun to eliminate the excess of antibody. Two additional washes were carried out. The cell pellet was finally resuspended in a 1:300 dilution of mouse anti-human IgG1 Fc-PE (Ref.: 9054-09, Southern Biotech) and a 1:500 dilution of the Brilliant Violet 421™ anti-human EGFR Antibody (Ref.: 352911, BioLegend) in PBS-BSA. Samples were then washed, and labeled cells were analyzed on a FACSCalibur or FACSCanto II flow cytometer (Becton-Dickinson), and the data were analyzed with FlowJo software (BD). For multiplexing, the following antibodies were used (all from Cytognos, S.L.): FITC-labeled anti-IgG1, PE-labeled anti-IgG2, APC-labeled anti-IgG3, APCC750 labeled anti-IgG4, PEcy7-labeled anti-IgM, and PerCPcy5.5-labeled anti-IgA (for IgA1 and IgA2).

### Automatic classification of samples

Flow cytometry data were saved as Flow Cytometry Standard (FCS) files and imported into R (R Core Team, 2013) with the read.FCS function of the flowCore (Ellis *et al*, 2017) R library. For selecting the main cell population and removing death or aggregated cells, an automatic gating over de SSC.A and FSC.A channels was done with a robust model-based clustering using a t-mixture model with Box–

Cox transformation as implemented in the flowClust R library (Lo *et al*, 2009). Selected cell population signals in FL2.A and FL7.A channels was modeled as a multivariate normal distribution using an expectation maximization algorithm for fitting mixtures of multivariate skew normal distributions as implemented in the EMMIXskew library (Wang *et al*, 2009). Our Score for classifying the samples as positive or negatives was the $\sigma$(FL2.A,FL7.A) of the co-variance matrix defining the multivariate normal distribution. For fine-tuning of the cutoff point of the co-variance for the classification of samples, we have used a set of training samples composed of negative samples (pre-COVID-19), and samples pre-classified as positive or negative based on alternative tests such as ELISA or PCR. Additionally, we have enriched this pool with serial dilutions of clearly positive samples, until no signal was observed. In total, we have used 168 flow cytometry readings, which once ordered according to the co-variance of the FL2.A and FL7.A channels, were splitted into positive and negative classes, without error, using 0.024 as the cutoff point. Further on, we have used this threshold to classify remaining samples, considering as negative those below the threshold.

### Commercial kits

The following commercial serological tests were assayed:

- Feal Test of Hangzhou Alltest Biotech, reference number RPP25COV1925.
- Sienna Test of Salofa Oy, reference number 102221.
- BiosSynex Test of Biosynex Swiss, reference number SW40005.
- Vircell ELISA test of Vircell SL, reference number G1032.
- COVID-19 VIRCLIA IgM + IgA monotest, Vircell SL, reference number VCM098.

### ELISA

96-well plates (MaxiSorp NUNC-Immunoplate) were coated overnight at 4°C with S1 (2 µg/ml) and were subsequently blocked for 1 h with 1% BSA (Sigma). Coated plates were incubated with the diluted sera for 1 h at room temperature. Bound antibodies were detected by incubation with mouse anti-human IgG1 secondary antibody coupled to horseradish peroxidase (HRP; Southern Biotech) diluted 1/6,000 in 1% BSA in PBS which was then detected using an ABTS substrate solution (Invitrogen). The OD at 415 nm was determined on a iMark microplate reader (Bio-Rad). Serum from a healthy individual previous to COVID-19 was used as negative control.

### Immunoprecipitation and Western blot

For surface biotinylation, $20 \times 10^6$ Jurkat-S cells were incubated for 45 min on ice with 0.5 mg/ml of sulfo-NHS-biotin (Pierce) in PBS supplemented with 0.1 mM $CaCl_2$ and 1 mM $MgCl_2$. After washing, the cells were lysed in Brij96 lysis buffer (0.33% Brij96, 150 mM NaCl; 20 mM Tris–HCl pH 7.8; 10 mM iodoacetamide; 1 mM PMSF; 1 µg/ml aprotinin; 1 µg/ml leupeptin) and immunoprecipitation was carried out with a mix of protein G beads and anti-T2A antibody (anti-2A peptide, clone 3H4 monoclonal antibody cat #MABS2005, EMD millipore) and incubated overnight

with rotation at 4°C. The bead-bound material was washed 5 times and subjected to SDS–PAGE. The proteins recovered were then transferred to a nitrocellulose membrane that was probed with streptavidin–horseradish peroxidase (Southern Biotech), diluted 1:20,000. Binding of streptavidin–peroxidase was then visualized using ECL(Pierce).

### Neutralization assay with pseudotyped virus

Lentiviral supernatants were produced from transfected HEK-293T cells as described previously (Martínez-Martín et al, 2009). Briefly, lentiviruses were obtained by co-transfecting plasmids psPAX2 (gag/pol), pHRSIN-GFP, and either a truncated S envelope (pCR3.1-St) (Robbiani et al, 2020) or VSV envelope (pMD2.G) using the JetPEI transfection reagent (Polyplus Transfection). Viral supernatants were obtained after 48 h of transfection. Polybrene (8 μg/ml) was added to the viral supernatants prior to transduction of $ACE2^+HEK293T$ cells. A total of $15 \times 10^3$ $ACE2^+$ HEK293T cells per well in a 96-well plate were seeded the day before transduction. Serially diluted plasma was incubated with viral supernatant for 1 h at 37°C prior addition to the cells. Cells were centrifuged for 70 min at 1,600 $g$ and left in culture for 48 h and then were resuspended in PBS with 2% FBS and 5 mM EDTA and fixed with 2% paraformaldehyde. $GFP^+$ cells were then analyzed on a FACSCanto II flow cytometer (Becton-Dickinson), and the data were analyzed with FlowJo software (BD).

### Formation of syncytia

To analyze the formation of syncytia between Jurkat-S and HepG2 cells, HepG2 cells were detached from the culture plate by trypsinization and resuspended at a concentration of $3 \times 10^6$ cells/ml in PBS. Jurkat-S cells were collected by centrifugation and resuspended in PBS at the same concentration. HepG2 cells were labeled with Cell Trace Far Red (CTFR, Invitrogen) for 5 min at 37°C in PBS; Jurkat-S were labeled with CFDA-SE (CFSE, Invitrogen) under the same conditions. Both dyes were used at a final concentration of 5 μM. Labeling was stopped by adding complete medium and the cells washed with medium and finally mixed at different ratios in complete RPMI medium + 5% FBS and plated overnight at 37°C. Formation of syncytia was analyzed by flow cytometry by calculating the percentage of cells double positive for CFSE and CTFR. Syncytia formation was confirmed by confocal microscopy. Syncytia was selected by FACS sorting and allowed to bind to poly-L-lysine-coated glass for 30 min and fixed for 10 min at room temperature with 4% paraformaldehyde. Cell nuclei were stained with 1 μg/ml DAPI in PBS for 5 min, and finally, samples were mounted with Prolong Antifade (Molecular probes). Samples were observed by LSM 710 laser scanning confocal microscope. Image processing was performed using Zen software.

### Statistics

Unpaired two-tailed Student $t$-tests were used to compare statistical significance between two groups of MFI values that followed a normal distribution. A paired $t$-test was used to compare series of dilutions of two samples. All data were analyzed using the GraphPad Prism 7 software.

### The paper explained

**Problem**

Current serological tests for detection of antibodies to SARS-CoV-2 are based on the use of recombinant fragments of proteins, including the spike S protein, that do not reproduce the native form of the protein in the context of the full virus or infected cells. This may lead to missing important antibodies that recognize conformational epitopes present in the native forms.

**Results**

By expressing in a stable manner the S protein of SARS-CoV-2 and a EGFR reporter construct on the surface of the human T lymphoblastic leukemia cell line Jurkat, we have created a flow cytometry-based method of detection of antibodies against the S protein that results more sensitive, more specific, and more relevant for the detection of functional neutralizing antibodies than those using recombinant proteins.

**Impact**

The method will allow to assess in a precise manner the degree of humoral immunity in human populations before and after vaccination, allowing to evaluate how close we are to the sought herd immunity.

Serum samples were received coded from the providers, and the experimentalists were blinded to their nature until all data analysis was finalized. Sample analysis was carried out in duplicate or triplicate, and all experiments were repeated a minimum of two times.

## Data availability

This study includes no data deposited in external repositories. Jurkat-S cells are available for academic research upon request to B. Alarcón.

Expanded View for this article is available online.

## Acknowledgements

We are indebted to Valentina Blanco and Tania Gómez for their expert technical assistance. We thank Dr. Peter Cherepanov, Annachiara Rosa, and Chloe Roustan (Crick COVID-19 Consortium, Francis Crick Institute, London, UK) for a generous gift of recombinant SARS-CoV-2 S1 and RBD antigens. We also thank Dr. P. Beniasz, The Rockefeller University, for providing constructs, and all volunteers of the CBMSO for generously participating in the study. This work was funded by intramural grant CSIC-COVID19-004: 202020E081 (to B.A.) and CSIC-COVID19-004: 202020E165 (to MF). L.H has been supported by an FPI fellowship from the Spanish Ministry of Science and Innovation. I.B. has been supported by an H2020-MSCA-ITN-2016 training network grant of the European Union (GA 721358).

## Author contributions

LH and PD performed research and analyzed the data. DA analyzed data and created the algorithm for classification. IB helped with ELISAs and other experimentation. GC provided the original idea and edited the manuscript. MAL, PM-F, SS-V, and FS-M provided clinical samples and data. MF and HMvS

Lydia Horndler et al

*EMBO Molecular Medicine*

analyzed data and supervised research. BA supervised and designed research, analyzed data, and wrote the manuscript.

## Conflict of interest

The authors have issued a patent application owned by CSIC.

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
