## [Review Process File · EMBO Molecular Medicine]

FLOW CYTOMETRY MULTIPLEXED METHOD FOR THE DETECTION OF NEUTRALIZING HUMAN ANTIBODIES TO THE NATIVE SARS-CoV-2 SPIKE PROTEIN

Lydia Horndler, Ivaylo Balabanov, Georgina Cornish, Miguel Llamas, Sergio Serrano-Villar, Manuel Fresno, Hisse van Santen, Balbino Alarcon, David Abia, Pedro Martinez-Fleta, Francisco Sanchez-Madrid, and Pilar Delgado

DOI: [10.15252/emmm.202013549](https://doi.org/10.15252/emmm.202013549)

Corresponding author: Balbino Alarcon (balarcon@cbm.csic.es)

Review Timeline:

Submission Date:	7th Oct 20
Editorial Decision:	28th Oct 20
Revision Received:	4th Dec 20
Editorial Decision:	24th Dec 20
Revision Received:	13th Jan 21
Accepted:	15th Jan 21

Editor: Zeljko Durdevic

Transaction Report:

28th Oct 2020

Dear Prof. Alarcon,

Thank you for the submission of your manuscript to EMBO Molecular Medicine. We have now received feedback from the three reviewers who agreed to evaluate your manuscript. As you will see from the reports below, the referees acknowledge the interest of the study but also raise serious and partially overlapping concerns that should be addressed in the major revision of the current manuscript. Please consider formatting your manuscript as a report article as suggested by the referee #3. Please check "Author Guidelines" for more information.

<https://www.embopress.org/page/journal/17574684/authorguide#reportsarticleguide>

Addressing the reviewers' concerns in full will be necessary for further considering the manuscript in our journal, and acceptance of the manuscript will entail a second round of review. EMBO Molecular Medicine encourages a single round of revision only and therefore, acceptance or rejection of the manuscript will depend on the completeness of your responses included in the next, final version of the manuscript. For this reason, and to save you from any frustrations in the end, I would strongly advise against returning an incomplete revision.

I look forward to receiving your revised manuscript.

Yours sincerely,

Zeljko Durdevic

***** Reviewer's comments *****

Referee #1 (Comments on Novelty/Model System for Author):

In this manuscript the authors have presented a novel technique for the assessment of COVID-19 serology. The method is novel and the data are compelling. Consequently, I expect the potential impact to be significant, mostly as a research tool as opposed to a clinical assay because of the complexity and expense of flow cytometry versus ELISA. Nevertheless, I have suggested some additional studies that should significantly enhance the manuscript. The authors have failed to provide any analysis of false positives and false negatives which are essential for the assessment of the value of the findings. I do not believe asking for this type of analysis is excessive.

Referee #1 (Remarks for Author):

The authors have presented compelling data on an important and interesting topic, the serological evaluation for specific anti-SARS-CoV2 antibodies. The flow cytometric assay that they have developed demonstrates excellent sensitivity and flexibility for the detection of various immunoglobulin classes and subclasses. Potentially the assay has a place in the analysis of patient samples. However, the authors have failed to include some important findings. They need to include an analysis of the threshold for positivity. By collecting these data they could perform a ROC analysis. For instance, the analysis of 50 pre-COVID-19 sera could be used as a gold standard negative set, and the analysis of sera from 50 patients with PCR positive people 4-8 weeks after infection could be used as a gold standard positive set (screening to eliminate immunodeficient persons). Without this the analysis of the determinations of false positives and false negatives (sensitivity, specificity, positive predictive value, negative predictive value) are not possible.

Page 4, last paragraph: the authors refer to Jurkat as "human hematopoietic cells"; however, it is more accurate and preferable to refer to Jurkat as "human T lymphocytic tumor cells". Jurkat cells have no "hematopoietic" properties.

Page 5, first paragraph: the authors write, "This data suggest ..." which is internally inconsistent. It is correct to write, "These data suggest ...". Again, later in this paragraph, "This data ..." should be changed to "These data ...".

Table 1: the color coding is confusing. The authors indicate the meaning for cells in green, yellow, and pink, but there are lots of cells in a light blue color and there is no indication what that color signifies.

Figure 2, legend: the authors state that "negative values for the flow cytometry test are those with a S/EGFR MFI ratio lower than 0.5". How this threshold was determined is not clear from the manuscript.

Page 6, line 14: the authors have cited Figure 1d but they should have cited Figure 2d.

Figure 3: the data shown do not seem responsive to the comments in the text involving a comparison of the sensitivity of the flow cytometric assay for anti-S immunoglobulin versus the ELISA. The correlational analysis in Figure 3b seems to show that divergence in sera with some showing a relatively flat regression and other showing a relatively steep regression. However, the authors state, "It is clear that detecting S-specific IgG1 using the Jurkat-S FC assay increases sensitivity for detecting SARS-CoV2-exposure in individuals testing negative by ELISA."; however, that conclusion is not obvious since the authors do not show a threshold for positivity. An analysis of pre-COVID, definitively negative sera would show the variance in values for all 3 assays. This determination is missing in this figure. Figure 3c does demonstrate an important analysis but again it appears only a single pre-COVID negative serum was included. It is essential to assess the variance in this analysis to set a threshold for positivity.

Figure 3c: the authors state that "all sera, including that of donor #58, were clearly positive by FC even at a 1:450 dilution"; however, the data presented do not support that contention. The data in the figure are too closely displayed to be certain but most importantly, there is no cutpoint for positivity shown in the figure.

Figure 2 and Figure 4: the donors indicated as RYC are not explained in the text until page 12, well after the description of the data in these 2 figures. The speculation about donor RyC65 on page 8 is inappropriate since no data are presented indicating that the donor was infected at all. This

speculation also appears on page 9. How do the authors know that the 3 cases are actually positive? They only tested positive in the flow assay and not in any other assay including PCR. Subsequent follow-up with these donors may elicit a history of eventual infection. Without that verification, it is not certain that the flow assay positives are not false positives.

In the discussion the authors indicate that the flow method is disadvantageous because of the requirement for a flow cytometer. They go on to explain that flow cytometers are prevalent. They do not discuss the expense of running a flow cytometric assay or the degree of complexity of flow cytometry compared to ELISA.

Referee #2 (Comments on Novelty/Model System for Author):

Technical quality: The study appears very well executed and the tests are described in sufficient detail to allow repetition.

Novelty: The study has been performed with adherent HEK cells, the advantage with Jurkat cells is that they are non-adherent.

Medical impact: Although as the authors argue many hospital laboratories have advanced flow cytometry equipment, this is not high throughput. So studies as this may be used to quantify the level of false negative test with the more high throughput tests.

Adequacy of the model: the full length spike proteins appears to display conformational epitopes which are difficult to capture.

Referee #2 (Remarks for Author):

In this paper the authors describe a sensitive and quantitative flow cytometry method using Jurkat cells transfected to stably express the full-length native spike of SARS-CoV-2. They show that antibodies can be detected in individuals regardless of the result of other tests.

Technical quality: The study appears very well executed and the tests are described in sufficient detail to allow repetition.

Novelty: The study has been performed with adherent HEK cells, the advantage with Jurkat cells is that they are non-adherent.

Medical impact: Although as the authors argue many hospital laboratories have advanced flow cytometry equipment, this is not high throughput. So studies as this may be used to quantify the level of false negative test with the more high throughput tests.

Adequacy of the model: the full length spike proteins appears to display conformational epitopes which are difficult to capture otherwise.

Major comment: The authors display data for a limited number of Covid19 patients and asymptomatic individuals. I would clearly have expected a higher number of negative sera to qualify whether this method reliably can differentiate "true positive" and "true negative". The Jurkat cell line is a cancer cell which may have upregulated cancer associated proteins that may be recognized by antibodies.

Referee #3 (Comments on Novelty/Model System for Author):

The system presented may provide higher robustness than existing flow cytometry approaches,

because of the internally normalized measurements through a mono-cystronic self-cleaving SARS-CoV-2 Spike protein/human EGFR, and because the Jurkat-S system is amenable to standardization because of its stably-transfected nature and easier handling of cells growing in suspension.

Referee #3 (Remarks for Author):

Major points

This article presents a Flow cytometry ratiometric method for the detection of neutralizing serum antibodies against the SARS-CoV-2 Spike protein. The main claims are high specificity and sensitivity, and overall better performance than the tested ELISA procedures. An additional claim is that of higher robustness than existing flow cytometry approaches, because of the internally normalized measurements through a mono-cystronic self-cleaving SARS-CoV-2 Spike protein/human EGFR, and because the Jurkat-S system is amenable to standardization because of its stably-transfected nature and easier handling of cells growing in suspension.

This article has merits, but shows weaknesses in its validation design and in its field-testing outcomes.

1. "recombinant fragments of S miss the quaternary structure of the S protein trimer, which is the native form of the spike protein in the viral envelope. Therefore, possible neutralizing antibodies directed against the native S trimer could be missed in serological tests based on the expression of recombinant proteins."

This point is key to the paper, and fundamental for more effective diagnostics. Please provide supporting evidence, both for native versus non-native configuration of the S protein trimer and for loss of detection of bona fide anti-corona virus serum Ig from COVID-19 patients when using recombinant fragments of S.

2. The presentation of the key findings of the article in Table 1 is confusing.

- All patients must have scored positive for the presence of the virus RNA at some point or another, and this needs to be indicated.

- viral load at the time of first positivity in the PCR diagnostics has to be reported.

Patients then need to be grouped and presented by:

- disease severity,

vs

- asymptomatic cases.

3. Sixty six cases and 30 controls are too few. A validation case series need to be added to this training series.

4. Serum positivity needs to be sequentially assessed along follow-up, in particular from disease onset to disease waning.

An interesting issue is if longer-lasting disease may lead to higher serological response.

Conversely, age may associate as a continuum variable to lower responses.

5. As mentioned by the authors, at least two other flow cytometry assays for detecting Ig in the serum of COVID patients have been developed. In the hands of Ng et al. (Ng, K. et al. Preprint at bioRxiv <https://doi.org/10.1101/2020.05.14.095414>) flow cytometry was more sensitive than ELISA in detecting anti-SARS-CoV-2 Ig. In the case of Grzelak L et al. flow cytometry showed an up to 3% false-positive rate (Science Translational Medicine 2020; 12(559): eabc3103).

The performance of the proposed flow cytometric procedure versus ELISA needs to be formally assessed, and percent sensitivity and specificity need to be determined in comparative assessments of bona fide positive and negative control cases, independent from the study cohort.

5. Positivity threshold: this needs to be established by objective quantitative methods, with statistical robustness.

A near-zero background staining must be reached, for robust translation into clinical diagnostics.

6. Results: "serum from donor #48 and, to a lesser extent, from donors #8 and #49, were also able to neutralize the S protein pseudotyped lentivirus (Fig. 4a), suggesting that these serum samples contain neutralizing antibodies despite being seronegative by ELISA (Fig. 3b and 3c). These data show that the Jurkat-S FC assay can be superior to ELISA for detecting protective immunity to SARS-CoV-2."

Titration of neutralizing activity, as an independent quantification method for amounts of Ig, is required to independently validate Flow cytometry versus ELISA.

More frequent instances of "detection" by flow may simply indicate false-positives.

7. "Finally, the comparison of Absorbance values in the two ELISA tests (anti-S1 and anti-RBD) produced a good-fitted straight line, whereas the comparison of the FC MFI with the absorbance values (against S1 and RBD) does not adjust to a straight line (Figure 3b)."

This is worrisome. Linear correspondence is expected in the case of reliable quantitation. It is the experience of this reviewer that instances on non-linear signals versus progressive dilutions stem from saturation of specific binding sites at the highest concentrations employed. Serial serum dilutions would provide multiple measurements per individual patient and per sequential serum drawings.

Additional issues

a. Results: "After overnight incubation, we detected a Jurkat-S dose-dependent formation of mixed-cell syncytia"

Flow cytometry data indicate the formation of aggregates. Is there any evidence for the formation of syncytia via cell-cell fusion? Immunofluorescence microscopy analysis is required, whereby mixing of the two cytoplasmic labels would be expected.

A fluorescence energy transfer test could also be utilized in flow cytometry, using appropriate pairs of soluble cytoplasmic fluorophores, with excitation of the donor fluorophore and collection of

emission from the acceptor fluorophore.

b. Discussion: "A FC-based method has been previously described using HEK293T cells that overexpress the S protein. Compared to this, the Jurkat-S system described here offers the advantage of employing a non-adherent cell line that does not require methods such as trypsinization to place them in suspension."

HEK-293T, may not require trypsin, and can be pipetted out from cell culture plates (Grzelak L. Science Translational Medicine 2020; 12(559): eabc3103). HEK-293T typically offer rather high transient transfection efficiency. The stably-transfected Jurkat offers potential advantages as for reproducibility, but this requires comparison with the HEK-293T system, and lower variance of expression levels over time in culture versus HEK-293T needs to be shown.

c. Fig. 1C, D: WB show excessive background or staining in the negative control, and have to be repeated.

Overall, the manuscript would be better suited to publication as a short report in EMBO Molecular Medicine.

Referee #1 (Comments on Novelty/Model System for Author):

In this manuscript the authors have presented a novel technique for the assessment of COVID-19 serology. The method is novel and the data are compelling. Consequently, I expect the potential impact to be significant, mostly as a research tool as opposed to a clinical assay because of the complexity and expense of flow cytometry versus ELISA. Nevertheless, I have suggested some additional studies that should significantly enhance the manuscript. The authors have failed to provide any analysis of false positives and false negatives which are essential for the assessment of the value of the findings. I do not believe asking for this type of analysis is excessive.

We appreciate the positive and constructive comments of the Referee. We think that thanks to his/her comments we have greatly improved the manuscript.

Referee #1 (Remarks for Author):

The authors have presented compelling data on an important and interesting topic, the serological evaluation for specific anti-SARS-CoV2 antibodies. The flow cytometric assay that they have developed demonstrates excellent sensitivity and flexibility for the detection of various immunoglobulin classes and subclasses. Potentially the assay has a place in the analysis of patient samples. However, the authors have failed to include some important findings. They need to include an analysis of the threshold for positivity. By collecting these data they could perform a ROC analysis. For instance, the analysis of 50 pre-COVID-19 sera could be used as a gold standard negative set, and the analysis of sera from 50 patients with PCR positive people 4-8 weeks after infection could be used as a gold standard positive set (screening to eliminate immunodeficient persons). Without this the analysis of the determinations of false positives and false negatives (sensitivity, specificity, positive predictive value, negative predictive value) are not possible.

We thank the Referee for this suggestion which we agree was necessary to validate the FC test. We have now analyzed 52 samples from the Hospital de la Princesa in Madrid most of them analyzed by PCR and by other serological method. With this set of samples we find a perfect correlation between serum samples from previously identified as PCR+ patients and our test (>97.5%). There was only one discrepancy with one sample from a PCR+ patient (HUP58) that was identified as seronegative by the ELISA method and now as seronegative in our FC method. Furthermore, we show that unlike other serum samples, the HUP59 serum did not neutralize in our assay with pseudotyped lentivirus (Fig. 5C). Thus, we think we can conclude that the sample corresponds to a false PCR+.

In addition, we have 52 sera from pre-COVID donors and identified all of them as seronegative in our FC assay. With this result, we could claim a rate of false seropositive close to 0%. All this new data are shown in Figure 5 and Table EV3

Page 4, last paragraph: the authors refer to Jurkat as "human hematopoietic cells"; however, it is more accurate and preferable to refer to Jurkat as "human T lymphocytic tumor cells". Jurkat cells have no "hematopoietic" properties.

Correct! We now say: ...using stably transfected Jurkat, a human leukemic T cell line, that co-express...

Page 5, first paragraph: the authors write, "This data suggest ..." which is internally inconsistent. It is correct to write, "These data suggest ...". Again, later in this paragraph, "This

data ..." should be changed to "These data ...".

We have made those changes in the Text according to the Referee's suggestions.

Table 1: the color coding is confusing. The authors indicate the meaning for cells in green, yellow, and pink, but there are lots of cells in a light blue color and there is no indication what that color signifies.

We apologize for the confusion. We have now replaced the light blue color by green in all Tables in order to make clear that we indicate positive sera.

Figure 2, legend: the authors state that "negative values for the flow cytometry test are those with a S/EGFR MFI ratio lower than 0.5". How this threshold was determined is not clear from the manuscript.

We now explain it in the legend to Figure 2B: "This ratio was set in order to fit most of the data negative for the other serological tests (pink triangles) under that threshold".

Nonetheless, this threshold has been now replaced by another one that derives from a Score calculated according to the slope of S/EGFR MFI data and shape of the curves as explained in Methods and Fig. EV1. The new threshold defines clearly the limit between positive and negative samples as illustrated in Figure 7C.

Page 6, line 14: the authors have cited Figure 1d but they should have cited Figure 2d.

This is now clarified. We refer now to Fig. 4E and Fig. 1F.

Figure 3: the data shown do not seem responsive to the comments in the text involving a comparison of the sensitivity of the flow cytometric assay for anti-S immunoglobulin versus the ELISA. The correlational analysis in Figure 3b seems to show that divergence in sera with some showing a relatively flat regression and other showing a relatively steep regression. However, the authors state, "It is clear that detecting S-specific IgG1 using the Jurkat-S FC assay increases sensitivity for detecting SARS-CoV2-exposure in individuals testing negative by ELISA."; however, that conclusion is not obvious since the authors do not show a threshold for positivity. An analysis of pre-COVID, definitively negative sera would show the variance in values for all 3 assays. This determination is missing in this figure. Figure 3c does demonstrate an important analysis but again it appears only a single pre-COVID negative serum was included. It is essential to assess the variance in this analysis to set a threshold for positivity.

We have now included a cohort of 52 pre-COVID samples to determine the specificity of the FC method, in addition to samples from PCR+ donors. All this is in Figure 5. In addition, we have now developed a Score based on slope and shape of the 2D anti-S/anti-EGFR plots, providing a clear positive/negative discrimination (Methods, Fig. 4F, Fig. EV1, Fig. 6B and Fig. 7C). The samples of Table EV1 and Fig. 3 have been reanalyzed with the newly developed Scoring system. The reanalysis confirms our claims for Fig. 3. However, in order to maintain the narrative structure, we have preferred to show the data as they were in the original version and illustrate the progressive development of the method with new Figures and examples.

Figure 3c: the authors state that "all sera, including that of donor #58, were clearly positive by FC even at a 1:450 dilution"; however, the data presented do not support that contention. The data in the figure are too closely displayed to be certain but most importantly, there is no cutpoint for positivity shown in the figure.

The cutpoint in the Figure is established by the pre-COVID-19 results. Although, as said in response to the previous question, all those data have been verified according to our new classification system, we have introduced a t-test to show that the differences between sera #8, #46, #48 and #49 by FC to the pre-COVID-19 sample at 1:450 dilution are significant.

Figure 2 and Figure 4: the donors indicated as RYC are not explained in the text until page 12, well after the description of the data in these 2 figures. The speculation about donor RyC65 on page 8 is inappropriate since no data are presented indicating that the donor was infected at all. This speculation also appears on page 9. How do the authors know that the 3 cases are actually positive? They only tested positive in the flow assay and not in any other assay including PCR. Subsequent follow-up with these donors may elicit a history of eventual infection. Without that verification, it is not certain that the flow assay positives are not false positives.

We have changed the order in which the RyC data is shown and is now illustrated in Figure 4 and discussed in the Text accordingly. We have also removed the speculation about a possible ongoing infection of donor RyC65, as suggested by the Reviewer. In regard to the number of RyC samples determined as positive, we have increased this number now to 5 (out of 30), after reanalysis using our new classification system. The Reviewer is right about the possibility that our data show false positives in contrast to previous ELISA and PCR data, which we claim are false negatives. However, the fact that those conflicting sera have neutralizing capacity (Fig. 4G) reinforces the FC data indicating that those sera are positive.

In the discussion the authors indicate that the flow method is disadvantageous because of the requirement for a flow cytometer. They go on to explain that flow cytometers are prevalent. They do not discuss the expense of running a flow cytometric assay or the degree of complexity of flow cytometry compared to ELISA.

We do not know what could be the cost of running the FC assay versus an ELISA. There are many factors that we ignore. For instance, the cost of producing recombinant proteins versus growing Jurkat cells in culture, the cost of the anti-EGFR monoclonal antibody, etc. So, we think we should not speculate further about these issues in the Discussion section. In regard to complexity, personnel trained in the Flow cytometer facilities of hospitals should have no problem to run the assay. However, all this is just a matter of commercialization of the product. We just wished to point out that the FC assay has to be carried out in specialized facilities.

Referee #2 (Remarks for Author):

In this paper the authors describe a sensitive and quantitative flow cytometry method using Jurkat cells transfected to stably express the full-length native spike of SARS-CoV-2. They show that antibodies can be detected in individuals regardless of the result of other tests.

Technical quality: The study appears very well executed and the tests are described in sufficient detail to allow repetition.

Novelty: The study has been performed with adherent HEK cells, the advantage with Jurkat cells is that they are non-adherent.

Medical impact: Although as the authors argue many hospital laboratories have advanced flow cytometry equipment, this is not high throughput. So studies as this may be used to quantify the level of false negative test with the more high throughput tests.

Adequacy of the model: the full length spike proteins appears to display conformational epitopes which are difficult to capture otherwise.

We appreciate the positive and constructive comments of the Referee. We think that thanks to his/her comments we have greatly improved the manuscript.

Major comment: The authors display data for a limited number of Covid19 patients and asymptomatic individuals. I would clearly have expected a higher number of negative sera to qualify whether this method reliably can differentiate "true positive" and "true negative". The Jurkat cell line is a cancer cell which may have upregulated cancer associated proteins that may be recognized by antibodies.

We have followed the recommendation of the Reviewer and have now analyzed a cohort of well-characterized samples from Hospital de la Princesa, with 40 "true positive" sera from patients tested by PCR and ELISA. In addition, we have included 52 additional pre-COVID sera as "true negative". All those data are in Figure 5 and show that the FC method is accurate with <2.5% false negatives and <2% false positives.

In regard to the possible presence of antibodies to cancer-associated proteins in the sera of patients, we have a control of parental Jurkat cells (i.e. not expressing S protein) which are perfect to exclude that possibility. So far, we have not found sera reactive with Jurkat-S AND parental Jurkat

Referee #3 (Comments on Novelty/Model System for Author):

The system presented may provide higher robustness than existing flow cytometry approaches, because of the internally normalized measurements through a mono-cystronic self-cleaving SARS-CoV-2 Spike protein/human EGFR, and because the Jurkat-S system is amenable to standardization because of its stably-transfected nature and easier handling of cells growing in suspension.

We appreciate the positive and constructive comments of the Referee. We think that thanks to his/her comments we have greatly improved the manuscript.

Referee #3 (Remarks for Author):

Major points

This article presents a Flow cytometry ratiometric method for the detection of neutralizing serum antibodies against the SARS-CoV-2 Spike protein. The main claims are high specificity and sensitivity, and overall better performance than the tested ELISA procedures. An additional claim is that of higher robustness than existing flow cytometry approaches, because of the internally normalized measurements through a mono-cystronic self-cleaving SARS-CoV-2 Spike protein/human EGFR, and because the Jurkat-S system is amenable to standardization because of its stably-transfected nature and easier handling of cells growing in suspension.

This article has merits, but shows weaknesses in its validation design and in its field-testing outcomes.

1. "recombinant fragments of S miss the quaternary structure of the S protein trimer, which is the native form of the spike protein in the viral envelope. Therefore, possible neutralizing antibodies directed against the native S trimer could be missed in serological tests based on the expression of recombinant proteins."

This point is key to the paper, and fundamental for more effective diagnostics. Please provide supporting evidence, both for native versus non-native configuration of the S protein trimer and for loss of detection of bona fide anti-corona virus serum Ig from COVID-19 patients when using recombinant fragments of S.

We have reinforced data demonstrating the native configuration of the S protein in the plasma membrane of Jurkat-S cells by showing that double-positive cells (Jurkat-S plus HepG2) in the flow cytometry data (Fig. 1D) are not doublets but represent real syncytia (new Fig. 1E). Since the S protein can mediate the fusion of Jurkat with ACE2+ cells, the S protein must be in native conformation which is in the form of trimers (as shown in Fig. 1C).

We cannot show that ELISA systems based on fragments of the S protein miss antibodies directed against the trimer or other parts of the protein (e.g. S2) not included in the construct. However, the discrepancy between our data and ELISA tests carried out at a diagnostics company (Table EV1) at Hospital Ramón y Cajal (Table EV2) and in different locations to personnel working at the CBMSO (Table EV4 and Fig. 7A) and the fact that antibodies detected as positive by FC and negative by ELISA have neutralizing activity (Fig. 4G, 5C, 6B and 7B) suggest that recombinant protein-based tests miss important antibodies with functional activity.

2. The presentation of the key findings of the article in Table 1 is confusing.

- All patients must have scored positive for the presence of the virus RNA at some point or another, and this needs to be indicated.

- viral load at the time of first positivity in the PCR diagnostics has to be reported.

Patients then need to be grouped and presented by:

- disease severity,

vs

- asymptomatic cases.

Unfortunately, we did not have information about PCR tests for most donors of Table EV1. We have now corrected this defect by including new cohorts of samples from Hospital de la Princesa. Those samples have been carefully monitored and described elsewhere (Ref. 9). The results are now presented in Fig. 5.

The classification of patients according to disease severity is shown in Fig. 3A.

3. Sixty six cases and 30 controls are too few. A validation case series need to be added to this training series.

We have now added 52 additional samples from Hospital de la Princesa and 52 pre-COVID samples (Fig. 5). In addition, we have now included 415 samples from volunteers working at the CBMSO

4. Serum positivity needs to be sequentially assessed along follow-up, in particular from disease onset to disease waning.

Unfortunately, we do not have access to samples taken at disease onset until disease waning. Besides, we would not expect to have reliable anti-S IgG1 until two weeks after disease onset. Nevertheless, in line with the suggestion of the Reviewer we have now included a small study on the detection of anti-S antibodies for several donors between June and October 2020 (Fig. 6A).

An interesting issue is if longer-lasting disease may lead to higher serological response. Conversely, age may associate as a continuum variable to lower responses.

We agree with the Reviewer that the association between disease severity, and therefore with days of active virus replication, and antibody titer is interesting. More or less that association is implicit in the Flow Cytometry plot of Fig. 3A. However, since this fact has been described previously (e.g. Chen et al, Signal Transduction Target Therapy doi: 10.1038/s41392-020-00301-9; Chen et al, Biomed Pharmacother 10.1016/j.biopha.2020.110629.), we did not wish to address this issue in the manuscript.

In regard to the possible inverse association with age, we do not see a clear pattern. For instance, if we plot data from Table EV1 (below) such association is not evident.

5. As mentioned by the authors, at least two other flow cytometry assays for detecting Ig in the serum of COVID patients have been developed. In the hands of Ng et al. (Ng, K. et al. Preprint at bioRxiv <https://doi.org/10.1101/2020.05.14.095414>) flow cytometry was more sensitive than ELISA in detecting anti-SARS-CoV-2 Ig. In the case of Grzelak L et al. flow cytometry showed an up to 3% false-positive rate (Science Translational Medicine 2020; 12(559): eabc3103).

With the data of Fig. 5 we can now say that the rate of false negatives in the FC Jurkat-S method is lower than 2.5% and the rate of false positives is lower than 2%.

The performance of the proposed flow cytometric procedure versus ELISA needs to be formally assessed, and percent sensitivity and specificity need to be determined in comparative assessments of bona fide positive and negative control cases, independent from the study cohort.

The ELISA test of Fig. 3 is homemade. However, all cohorts included in the revised version of the manuscript contain data on the result of commercial tests based on ELISA. We have found discrepancies with the FC Jurkat-S test in all cases, except the HUP cohort (Table EV3) probably because the selected samples were carefully chosen by two of the new co-authors to be clear cut. We have backed with neutralization data the FC data indicating that the discrepancies were due to ELISA false negatives.

5. Positivity threshold: this needs to be established by objective quantitative methods, with statistical robustness.

A near-zero background staining must be reached, for robust translation into clinical diagnostics.

Following the Reviewer's recommendation, we have now established an algorithm based on the slope of MFI for S staining vs EGFR staining and the shape of the population distribution to determine a Score that clearly distinguish positive from negative samples. This threshold is set for a Score of 0.024 and is shown in Figures 4, 5 and 7. The algorithm is described in Methods and explained in Fig. EV1. This method is automatic and can directly interpret data as it is provided by the flow cytometer.

6. Results: "serum from donor #48 and, to a lesser extent, from donors #8 and #49, were also able to neutralize the S protein pseudotyped lentivirus (Fig. 4a), suggesting that these serum

samples contain neutralizing antibodies despite being seronegative by ELISA (Fig. 3b and 3c). These data show that the Jurkat-S FC assay can be superior to ELISA for detecting protective immunity to SARS-CoV-2."

Titration of neutralizing activity, as an independent quantification method for amounts of Ig, is required to independently validate Flow cytometry versus ELISA. More frequent instances of "detection" by flow may simply indicate false-positives.

We have now carried out more neutralization tests using different dilutions to resolve conflicts between ELISA and FC data. Such results are in Fig. 4C, 5C and 7B.

7. "Finally, the comparison of Absorbance values in the two ELISA tests (anti-S1 and anti-RBD) produced a good-fitted straight line, whereas the comparison of the FC MFI with the absorbance values (against S1 and RBD) does not adjust to a straight line (Figure 3b)."

This is worrisome. Linear correspondence is expected in the case of reliable quantitation. It is the experience of this reviewer that instances on non-linear signals versus progressive dilutions stem from saturation of specific binding sites at the highest concentrations employed. Serial serum dilutions would provide multiple measurements per individual patient and per sequential serum drawings.

We show that the Absorbance data using S1 and the ELISA using the RBD fragments nicely fit to a linear distribution (Fig. 3B). However, comparison of FC MFI with Absorbance shows many outliers. The comparison between FC and ELISA data in Fig. 3B was done at a 1:50 dilution in all assays. The titration experiment of Fig. 3C shows that the discrepancies were not due to saturation of responses at 1:50 but that there are sera (#8, #46, #48 and #49) not detected by ELISA and detected by FC, independent of the dilution.

Additional issues

a. Results: "After overnight incubation, we detected a Jurkat-S dose-dependent formation of mixed-cell syncytia"

Flow cytometry data indicate the formation of aggregates. Is there any evidence for the formation of syncytia via cell-cell fusion? Immunofluorescence microscopy analysis is required, whereby mixing of the two cytoplasmic labels would be expected.

A fluorescence energy transfer test could also be utilized in flow cytometry, using appropriate pairs of soluble cytoplasmic fluorophores, with excitation of the donor fluorophore and collection of emission from the acceptor fluorophore.

We have followed the Reviewer's suggestion and included now immunofluorescence microscopy data to show that the CFSE and CTFR double-positive cells (Fig. 1D) are not cell doublets but syncytia in which both cells are totally fused (Fig. 1E). Our previous data showing that CFSE and CTFR double-positive cells were not detected when HepG2 cells were mixed with parental (S-negative) Jurkat cells (Fig. 1D) argued against the possibility of just counting cell doublets but, we agree, that the direct visualization by confocal microscopy (Fig. 1E) is decisive.

b. Discussion: "A FC-based method has been previously described using HEK293T cells that

overexpress the S protein. Compared to this, the Jurkat-S system described here offers the advantage of employing a non-adherent cell line that does not require methods such as trypsinization to place them in suspension."

HEK-293T, may not require trypsin, and can be pipetted out from cell culture plates (Grzelak L. Science Translational Medicine 2020; 12(559): eabc3103). HEK-293T typically offer rather high transient transfection efficiency. The stably-transfected Jurkat offers potential advantages as for reproducibility, but this requires comparison with the HEK-293T system, and lower variance of expression levels over time in culture versus HEK-293T needs to be shown.

Another advantage of the Jurkat-S system is the existence of an internal control (hEGFRt) that allows to normalize MFI values. In addition, the coordinated expression of S and hEGFRt allows to calculate a clear threshold value according to the slope and shape of the 2D plots (Fig. EV1). This allows an automatic classification of samples directly using flow cytometry data without human intervention. However, we prefer to leave the direct comparison the comparison with HEK-293T system to other scientists or for a future study.

c. Fig. 1C, D: WB show excessive background or staining in the negative control, and have to be repeated.

We have now repeated WB using a cleaner antibody for immunoprecipitation and the results show more clear now the presence of S protein monomer and multimers (up to trimer) on the surface of Jurkat-S cells. (Fig. 1B and 1C).

24th Dec 2020

Dear Prof. Alarcon,

Thank you for the submission of your revised manuscript to EMBO Molecular Medicine. I am pleased to inform you that we will be able to accept your manuscript pending the following final amendments:

1) With approaching holidays and the end of the year we encountered high number of submissions, so that our data editors were not able to process all received manuscripts before the holiday season. Therefore, we will send you the document with data editor's suggestions after the holidays and as soon as our data editors process your manuscript. Please do not submit your revised manuscript before we send you the file with data editor's suggestions. Thank you for your understanding.

2) Please address all the points raised by the referee #3.

***** Reviewer's comments *****

Referee #1 (Comments on Novelty/Model System for Author):

The authors have satisfactorily responded to suggestions from the first review. The manuscript is now appropriate for publication.

Referee #1 (Remarks for Author):

The authors have revised the manuscript to give it more impact and importance.

Referee #2 (Remarks for Author):

I think the authors sufficiently answered the reviewer comments and expanded the study.

Referee #3 (Remarks for Author):

Referee #3 (Remarks for the Authors):

We appreciate the improvements introduced in the revised text.

Among them, supporting findings for a native configuration of Jurkat-expressed of the SARS-CoV-2 Spike protein trimer, performance estimates on the ELISA assays and evidence on cell-cell fusion induced by the Jurkat transfectants.

Data presentation in Table 1 has been improved, additional data on control volunteers strengthen the determinations, as do data on waning of antibody titers as determined by the Jurkat assay.

Perhaps, the most significant improvement in the draft is the introduction of an algorithm for classifying measurements, via introducing non-arbitrary positivity thresholds.

Requests

1. Data on age vs antibody titers are of relevance (and encouraging), and should be added to the article.

2. According also to Referee # 1, please modify the first sentence of the Discussion, from: "Here we describe a method based on flow cytometry of a human cell line of hematopoietic origin that stably expresses the S protein of SARS-CoV-2"

to:

"Here we describe a method based on flow cytometry of a human T lymphoblastic leukemia cell line that stably expresses the S protein of SARS-CoV-2"

The authors performed the requested changes.

Reviewer #3

1. Data on age vs antibody titers are of relevance (and encouraging), and should be added to the article.

We guess that the Reviewer is referring to the lack of association between age and titer of antibody anti-S. Taking the S/EGFR MFI ratio from the seropositive individuals tested of the EMPIREO and HUP cohorts, it seems clear that older individuals (>60 year-old) are as competent as younger ones to make antibodies (see Figure below).

We have introduced a sentence in the second paragraph of page 13 (Discussion section) to point out to this observation:

“Another conclusion that can be drawn from the analysis of the Empireo and HUP cohorts (Tables 1 and 3) by the FC Jurkat-S method is that the humoral response to the S protein is not inferior for the older donors (ages 60-70 and more) than for younger ones, suggesting that age “per se” is not conditioning the response”

2. According also to Referee # 1, please modify the first sentence of the Discussion, from: "Here we describe a method based on flow cytometry of a human cell line of hematopoietic origin that stably expresses the S protein of SARS-CoV-2" to:

"Here we describe a method based on flow cytometry of a human T lymphoblastic leukemia cell line that stably expresses the S protein of SARS-CoV-2"

We have replaced the first sentence of page 12 (Discussion section) to follow exactly the Reviewer recommendation.

15th Jan 2021

Dear Prof. Alarcon,

We are pleased to inform you that your manuscript is accepted for publication.

Corresponding Author Name:

Journal Submitted to:

Manuscript Number: